# Retrieving cloud base height and geometric thickness using the oxygen A-band channel of GCOM-C/SGLI

Takashi M. Nagao[1], Kentaroh Suzuki[1], and Makoto Kuji[2]

[1]Atmosphere and Ocean Research Institute, The University of Tokyo, Kashiwa, Japan
[2]Nara Women's University, Nara, Japan

*Correspondence to*: Takashi M. Nagao (tmnagao@aori.u-tokyo.ac.jp)

**Abstract.** Measurements with a 763 nm channel, located within the oxygen A-band and equipped on the Second-generation Global Imager (SGLI) onboard the JAXA's Global Change Observation Mission – Climate (GCOM-C) satellite, have the potential to retrieve cloud base height (CBH) and cloud geometric thickness (CGT) through passive remote sensing. This study implemented an algorithm to retrieve the CBH using the SGLI 763 nm channel in combination with several other SGLI channels in the visible, shortwave infrared, and thermal infrared regions. In addition to CBH, the algorithm can simultaneously retrieve other key cloud properties, including cloud optical thickness (COT), cloud effective radius, ice COT fraction as the cloud thermodynamic phase, cloud top height (CTH), and CGT. Moreover, the algorithm can be seamlessly applied to global clouds comprised of liquid, ice, and mixed phases. The SGLI-retrieved CBH exhibited quantitative consistency with CBH data obtained from the ground-based ceilometer network, ship-borne ceilometer, satellite-borne radar and lidar observations, as evidenced by sufficiently high correlations and small biases. These results provide practical evidence that the retrieval of CBH is indeed possible using the SGLI 763 nm channel. Moreover, the results lend credence to the future use of SGLI CBH data, including the estimation of the surface downward longwave radiative flux from clouds. Nevertheless, issues remain that must be addressed to enhance the value of SGLI-derived cloud retrieval products. These include the bias of SGLI CTH related to cirrus clouds and the bias of SGLI CBH caused by multi-layer clouds.

## 1 Introduction

Cloud base height (CBH) and cloud top height (CTH) are fundamental properties that characterize the vertical extent of clouds and cloud radiative effects in the longwave (LW) spectrum. CBH is often combined with CTH to classify cloud types, which are crucial since they indicate specific weather conditions, atmospheric processes, and precipitation characteristics. Satellite CBH observations are essential for estimating the downward LW flux from clouds on a global scale, which cannot be measured directly from space. Therefore, reliable CBH data

obtained from satellite observations can reduce uncertainties in estimates of the downward LW flux in the Earth's radiation budget. This, in turn, can enhance our understanding of the atmosphere–surface energy flow, including latent heat transport associated with precipitation (Stephens et al., 2012).

Cloud geometric thickness (CGT), which is equal to the difference between CTH and CBH, is another crucial cloud property related to cloud microphysics and aerosol–cloud interactions. For example, the discrepancy

between the CGTs observed and those predicted by an adiabatic process model, based on the observed liquid water path or a pair of observed cloud optical thickness (COT) and cloud effective radius (CER) (Bennartz, 2007), serves as a coarse measure of adiabaticity affected by non-adiabatic processes such as drizzle formation and entrainment (Merk et al., 2016). Thus, observing CBH and CGT contributes to our understanding of the role of clouds in weather and climate through cloud microphysical processes.

Satellite-based cloud observing active instruments, cloud profiling radars such as the Cloud Profiling Radar (CPR) onboard the CloudSat mission (Stephens et al., 2008) and atmospheric lidars such as the Cloud–Aerosol Lidar with Orthogonal Polarization (CALIOP) onboard the Cloud–Aerosol Lidar and Infrared Pathfinder Satellite Observations (CALIPSO) mission (Winker et al., 2009), provide reliable measurements of the vertical distribution of clouds including CBH and CGT. However, they have several limitations. One limitation is

related to their ability to detect clouds. For instance, CloudSat/CPR has difficulty in detecting optically thin cloud layers associated with cirrus clouds. Similarly, CALIPSO/CALIOP has limitations in probing deeper optical depths in cloud layers. Nevertheless, the synergistic integration of radar and lidar profiles can mitigate this limitation and achieve a more detailed vertical resolution from the cloud base to the top, including multi-layer structures (Hagihara et al., 2010). Another limitation of these active sensors is that their measurements

are constrained to narrow nadir views along the satellite's orbit. This limitation is dependent on the technological development and cost of the instruments and persists to this day, including recently launched missions such as EarthCARE (Illingworth et al., 2015; Wehr et al. 2023).

Various attempts have been made to derive CBH and CGT using satellite-based passive instruments instead of active instruments. These methods can be classified into two types. The first approach involves inverse

estimation (retrieval) from multi-wavelength measurements of the oxygen absorption bands (including the oxygen A-band ranging from 759 to 771 nm and the oxygen B-band centered at 688 nm). This retrieval method has been implemented based on the channel configuration in the oxygen absorption bands: the reflectance spectrum in the oxygen A-band from the Orbiting Carbon Observatory-2 (Richardson et al., 2019; Yang et al.,

2021), two channels in the oxygen A-band of the third Polarization and Directionality of the Earth's Reflectances (POLDER-3) onboard the Polarization and Anisotropy of Reflectances for Atmospheric Sciences coupled with Observations from a Lidar (PARASOL) (Ferlay et al., 2010), a combination of the oxygen A-band channel and thermal infrared (TIR) channel of the Global Imager (GLI) on the Advanced Earth Observing Satellite – II (ADEOS-II) (Kuji and Nakajima, 2004; Kuji, 2013), and coincident measurements from two sensors providing the oxygen A-band and TIR channels, respectively: the Global Ozone Measurement Experiment (GOME) spectrometer and the Along Track Scanning Radiometer-2 (ATSR-2) on the European Remote-Sensing Satellite-2 (ERS-2) (Rozanov and Kokhanovsky, 2006); the SCanning Imaging Absorption spectroMeter for Atmospheric CHartographY (SCIAMACHY) and the Advanced Along-Track Scanning Radiometer (AATSR) onboard the European Environmental Satellite (Envisat) (Lelli and Vountas, 2018).

In addition to the literatures cited in the previous paragraph, there are several earlier studies that have investigated or attempted remote sensing of cloud geometric information using oxygen absorption channels while not retrieving both CBH and other geometric information (e.g. thickness). Examples include the use of oxygen A-band measurements (O'Brien and Mitchell, 1992), oxygen B-band measurements from the Global Ozone Monitoring Experiment (Desmons et al., 2019), and two channels in the oxygen A-band and B-band of the Earth Polychromatic Imaging Camera (EPIC) on the Deep Space Climate ObserVatoRy (DSCOVR) (Davis et al., 2018a; 2018b).

Another approach is to estimate other cloud properties correlated with CBH and CGT, such as CTH and COT, which are usually measured using passive instruments not equipped with oxygen absorption channels. This approach has been implemented using adiabatic (Seaman et al., 2017) or statistical models (Noh et al., 2017, 2022; Shao et al., 2023; Tan et al., 2023). Additionally, certain implementations have attempted to reconstruct vertical profiles, including CTH and CBH, as obtained with active instruments, using radiances or cloud properties observed with passive instruments (Barker et al., 2011; Okata et al., 2017; Leinonen et al., 2019).

Satellite-based passive instruments have several advantages and disadvantages compared to satellite-based active instruments for CBH and CGT observations. First, passive instruments enable the observation of the horizontal distribution of CBH and CGT with their wide swaths. Second, more passive instruments are launched and operated than active instruments. Additionally, passive instruments designed for cloud remote sensing typically have multi-wavelength channels, allowing for the retrieval of other fundamental cloud properties, such as COT and CER. However, despite these advantages, estimating CBH and CGT with an accuracy comparable to that of active remote sensing remains challenging, in principle, for passive remote sensing techniques.

In this study, we focused on the Second-generation Global Imager (SGLI) on the Global Change Observation Mission – Climate (GCOM-C) (Imaoka et al., 2010), which succeeded the ADEOS-II/GLI and was launched by the Japan Aerospace Exploration Agency (JAXA) at the end of 2017 (Tanaka et al., 2018). SGLI is a multispectral optical radiometer that has 19 spectral channels from 380 nm to 12 μm, including the oxygen A-band channel centered at 763 nm, with a wide swath of over 1,000 km and high spatial resolution ranging from

250 m to 1 km. The SGLI channel set offers a unique capability designed to acquire cloud properties essential for estimating the radiative flux components associated with clouds. Specifically, it includes shortwave infrared (SWIR) channels for retrieving COT and CER related to shortwave flux, TIR channels for retrieving CTH associated with upward LW flux from clouds, and an oxygen A-band channel for retrieving CBH related to downward LW flux from clouds. Currently, the SGLI operational cloud product provides retrievals of COT, CER, and CTH (Nakajima et al., 2019) as standard products; however, CBH and CGT have not yet been provided.

This study describes an algorithm to retrieve CBH and CGT data using the oxygen A-band channel of the SGLI and its validation against in-situ measurements with ceilometers and satellite-borne radar and lidar observations. The algorithm was implemented by coupling two algorithms: the four-channel algorithm originally introduced by Kuji and Nakajima (2001) and subsequently demonstrated with ADEOS-II/GLI measurements for water clouds (Kuji and Nakajima 2004; Kuji, 2013), which retrieves COT, CER, CTH, and CGT using the VNIR, SWIR, oxygen A-band, and TIR channels, respectively, and a cloud phase retrieval algorithm using multiple SWIR channels (Nagao and Suzuki, 2021). Through this coupling, this study aims to retrieve the CBH and CGT in a manner that combines other key cloud properties such that the SGLI's unique capability of multispectral measurements can be exploited for cloud remote sensing.

The remainder of this paper is organized as follows: Section 2 details the principles, implementation, and input data for the cloud retrieval algorithm using the SGLI oxygen A-band channel; Section 3 presents the results after applying the algorithm to the SGLI data; In Section 4, the CBH data derived from SGLI is validated against the CBH measured via ground-based and ship-borne ceilometers; In addition, the zonal means of CTH and CBH obtained from SGLI are validated against measurements obtained from a combined observation of CloudSat/CPR and CALIPSO/CALIOP; and Section 5 summarizes the findings and conclusions of this study and discusses the limitations and potential improvements of the algorithm.

## 2 Methods and data

This section provides comprehensive details on the principles (Sect 2.1), implementation (Sect 2.2), and input data (Sect 2.3) of the cloud retrieval algorithm that utilizes the SGLI oxygen A-band channel. A unique feature of our algorithm is that it combines the oxygen A-band channel with the VNIR, SWIR, and TIR channels, enabling the simultaneous retrieval of not only CTH and CBH (or CGT) but also other key cloud microphysical properties, specifically COT, CER, and the cloud thermodynamic phase.

### 2.1 Principles

The top-of-atmosphere (TOA) reflectance in the oxygen A-band channel (~763 nm), which is characterized as moderate to strong oxygen absorption, exhibits sensitivity to both CTH and CGT. This mechanism can be

described as follows. First, for simplicity, we assume a black surface with an albedo of zero, gas absorption only by oxygen, and a geometrically thin (CGT~0) but optically reasonably thick (COT > 0) plane-parallel cloud. Additionally, we assume that the cloud properties used to parameterize cloud scattering—namely, COT, CER, and cloud thermodynamic phase, are known. Under these conditions, the TOA reflectance in visible and near-infrared channels outside the oxygen absorption band (e.g., the VN9, VN11, and SW1 channels of SGLI, centered at 673 nm, 868 nm, and 1.05 μm, respectively) can be accurately represented using these cloud property parameters.

In the oxygen absorption channel, sunlight is significantly absorbed by the oxygen above the clouds before and after being reflected by the clouds on its path to the satellite. The TOA reflectivity in the oxygen absorption channel can be expressed with two additional parameters: CTH and the amount of oxygen above CTH. Conveniently, oxygen is well-mixed in the atmosphere, and its mixing ratio can be assumed to be globally constant and known. Thus, if the CTH (or cloud top pressure, equivalently) is given, the amount of oxygen above cloud can be immediately calculated. Therefore, when CGT~0, it is sufficient for parameterizing the TOA reflectance in the oxygen absorption channel to have CTH in addition to COT, CER, and cloud thermodynamic phase.

When CGT > 0, the CGT (or CBH) needs to be included as an additional parameter for explaining the TOA reflectance in the oxygen absorption band. When the cloud is geometrically thickened without changing the CTH (i.e., when CGT increases and CBH decreases), the TOA reflectance should decrease owing to oxygen absorption within the cloud layer between CTH and CBH. This is because, as the CBH decreases, sunlight travels a longer distance in the cloud layer, increasing the opportunities for oxygen absorption. These explanations provide an intuitive understanding of why the TOA reflectance in the oxygen absorption channel is sensitive to variations in both CTH and CGT (or CBH).

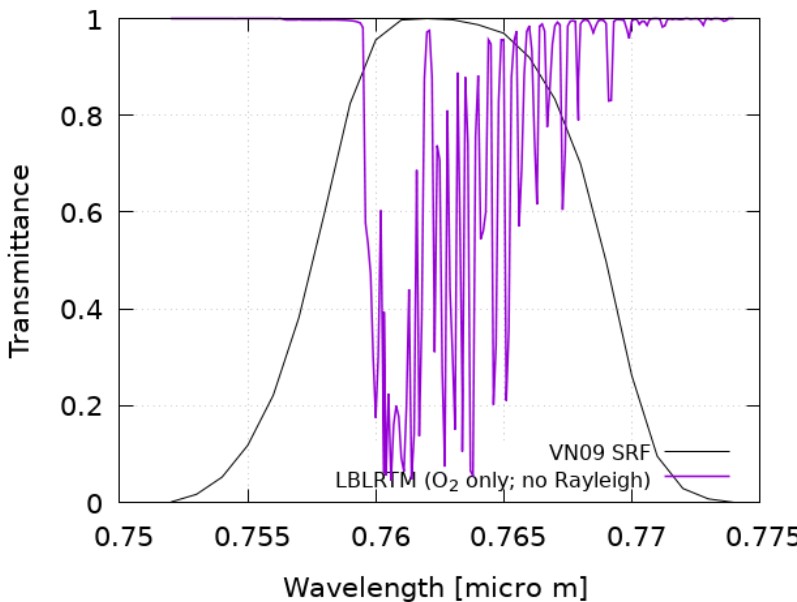

**Figure 1: Example of a transmittance spectrum in the oxygen A-band, simulated using the Line-By-Line Radiative Transfer Model (LBLRTM). The spectrum includes the spectral response function (SRF) of the Second-generation Global Imager (SGLI) oxygen A-band channel (VN9), centered at 763 nm.**

The retrievals of CTH and CGT from the oxygen absorption band through this principle can be facilitated by a pair of spectral channels with different sensitivities to these cloud parameters since the oxygen absorption channels are sensitive to both CTH and CGT. This is the case even when cloud properties other than CTH and CGT (i.e., COT, CER, and cloud thermodynamic phase) are known. Here, SGLI has only one oxygen A-band channel (called VN9), whose spectral response function is shown in Figure 1, alongside a simulated transmittance spectrum. As an alternative to a pair of the oxygen A-band channels, an approach using an oxygen A-band channel paired with a TIR channel highly sensitive to CTH has been proposed (Kuji and Nakajima, 2001). This approach can be implemented by using the SGLI TI1 (10.8 μm) as a TIR channel and VN9, as depicted in Figure 2, which illustrates the relationship between TOA radiance and reflectance at these channels for various specified CTH and CGT values.

Figure 2a illustrates the effective separation of CTH and CGT using SGLI TI1 and VN9. The "net" formed by TI1 and VN9 measurements, stretching along CTH and CGT, is regular and well-spread without overlap (multiple solutions) or distortion (excessive nonlinearity). This illustrates that CTH and CGT can be easily and uniquely determined from the given TI1 and VN9 measurements. The VN9 and TI1 measurements are also significantly dependent on other cloud properties, especially the COT, CER, and cloud thermodynamic phase (liquid, ice, or mixed). To determine CGT and CTH, at least three channels in VNIR and SWIR, in addition to VN9 and TI1, must be used simultaneously to retrieve these cloud properties. As detailed in the next section, this study implemented such simultaneous retrieval as a coupling of two algorithms: (i) the four-channel

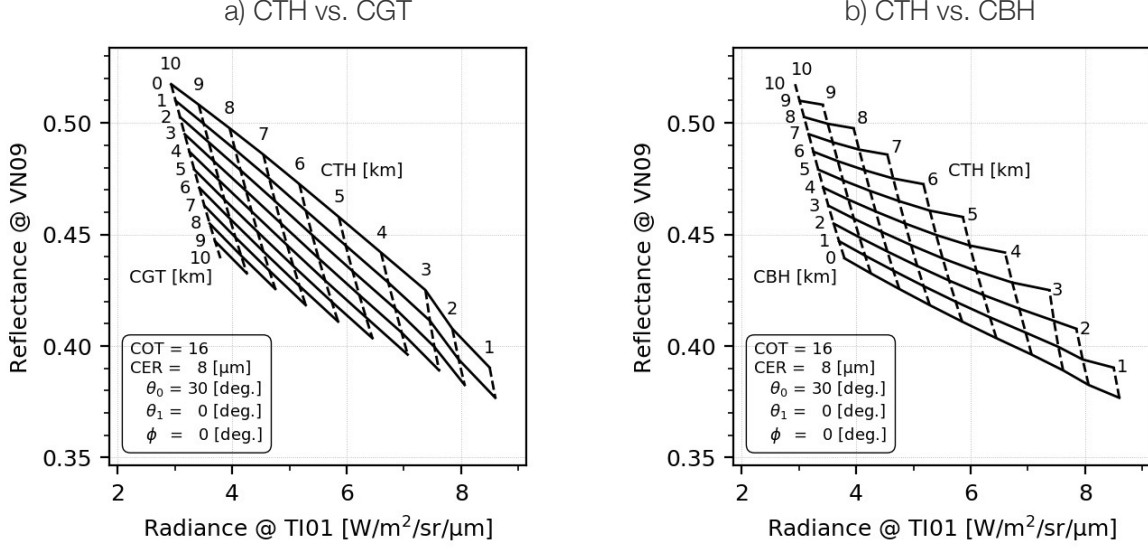

**Figure 2: a) Relationships between top-of-atmosphere radiance and reflectance at SGLI TI1 (10.8 μm) and VN9 (763 nm) channels, for various specified values of cloud top height (CTH) and cloud geometric thickness (CGT). b) Similar to a) but for cloud base height (CBH). COT, cloud optical thickness; CER, cloud effective radius; $\theta_0$, solar zenith angle; $\theta_1$, sensor zenith angle; $\phi$, relative azimuth angle.**

algorithm (Kuji and Nakajima, 2001) using the VNIR, SWIR, TIR, and oxygen A-band channels, and (ii) a cloud phase retrieval algorithm using multiple SWIR channels (Nagao and Suzuki, 2021).

Figure 2a implicitly constrains the direction of increasing CGT to correspond with a decreasing CBH. There are two possibilities for increasing CGT: increasing CTH or decreasing CBH. However, once CTH is
170 determined based on the TI1 measurement, the only information that can be extracted from the VN9 measurement is the CBH. Figure 2b depicts the CTH and CBH values in the VN9-TI1 measurement space. In this figure, the grid points are the same as those in Fig. 2a, although the shape of the net appears different due to the different manners in which the points are connected. Thus, Fig. 2b shows an alternative method for retrieving cloud geometrical parameters (CTH and CBH), equivalent to CTH and CGT in Fig. 2a, based on the
175 VN9 and TI1 measurements.

## 2.2 Implementation

### 2.2.1 Inverse estimation

This section describes the newly implemented CGT retrieval algorithm for the SGLI channels. As described
in the previous section, to determine CTH and CGT inversely from the SGLI TI1 and VN9 measurements, other cloud properties, specifically COT, CER, and the cloud thermodynamic phase, must be retrieved simultaneously. A promising approach for this task is the four-channel method developed by Kuji and Nakajima (2001). This

method can simultaneously retrieve COT, CER, CTH, and CGT using the VNIR, SWIR, TIR, and oxygen A-band channels, respectively. It has been demonstrated with ADEOS-II/GLI measurements for water clouds. The only drawback of this algorithm is that the thermodynamic phase of the cloud must be determined prior to retrieval. Conversely, a cloud phase retrieval algorithm using the three SWIR channels of SGLI (specifically, SW1, SW3, and SW4 centered at 1.05, 1.63, and 2.21 µm, respectively), developed by Nagao and Suzuki (2021), can simultaneously retrieve COT, CER, and the ice COT fraction (ICOTF). The ICOTF is a cloud phase representation defined as the fraction of ice COT to total COT (i.e., $ICOTF = COT_{ice}/(COT_{liquid} + COT_{ice})$). ICOTF values of zero and one indicate pure liquids and pure ice, respectively. In this study, we integrated the techniques of these two approaches to develop a novel algorithm. This integrated algorithm retrieved five cloud properties, COT, CER, ICOTF, CTH, and CGT, utilizing at least five channels, including the VNIR, SWIR, TIR, and oxygen A-band of SGLI.

Our algorithm employs the optimal estimation method framework (Rodgers, 2000), which seeks the optimal solution that minimizes the cost function $J$ that is given by the following equation:

$$J = [y - F(x)]^T S_e^{-1}[y - F(x)] + [x - x_a]^T S_a^{-1}[x - x_a], \qquad (1)$$

where $y$ and $F(x)$ represent the measurement vectors consisting of the measured and simulated TOA reflectances, respectively. $x$ represents the state vector, $x_a$ represents the a priori values for $x$, and $S_e$ and $S_a$ represent the covariance matrices for $y$ and $x_a$, respectively.

The iterative solution of the inverse problem through the Levenberg–Marquardt approach, based on the Gauss-Newton method for minimizing Equation (1), is determined using the following formula (Rodgers, 2000):

$$x_{i+1} = x_i + [(1 + \gamma) S_a^{-1} + K_i^T S_e^{-1} K_i]^{-1}\{K_i^T S_e^{-1}[y - F(x_i)] - S_a^{-1}[x_i - x_a]\}, \qquad (2)$$

where $x_i$ is the state vector at the $i$-th iteration, $K_i$ is the Jacobian matrix of $F(x)$ evaluated at $x_i$, $\gamma$ is a parameter chosen at each step of the iteration to reduce the cost function $J$.

In the analysis in Sects 3 and 4, we employed the seven SGLI channels, VN9 (763 nm), VN11 (868 nm), SW1 (1.05 µm), SW3 (1.63 µm), SW4 (2.21 µm), TI1 (10.8 µm), and TI2 (12.0 µm), to retrieve the five cloud properties. Hence, in this study, each variable in Equation (1) can be written as follows: $y = (R_{VN9}, R_{VN11}, R_{SW1}, R_{SW3}, R_{SW4}, I_{TI1}, I_{TI2})^T$ and $x = (COT, CER, ICOTF, CTH, CBH)^T$, where $R_\lambda$ and $I_\lambda$ are the measured TOA reflectance and radiances at the SGLI channels $\lambda$, respectively. Note that CBH, instead of CGT, was used as the element of $x$; however, as discussed in the previous section, CBH and CGT were equivalent in our implementation. The diagonal elements of $S_e$, consisting of the uncertainties in the TOA measurements, were obtained from the post-launch calibration information of the GCOM-C mission, while the

non-diagonal elements of $S_e$ were set to zero. In addition, $x_a$ and the diagonal elements of $S_a$ were given for the values summarized in Table 1, while the non-diagonal elements of $S_a$ were set to zero. As shown in Table 1, the lower limit of CBH was constrained by the greater value between the lifted condensation level (LCL) and the surface elevation. Note that the values in Table 1 could be assigned more appropriate prior distributions (mean, standard deviation, and even covariance) by using cloud property products from other satellite observations. However, since this is the first application of our algorithm to GCOM-C/SGLI, we used a normal distribution for simplicity with means of typical orders of magnitude and fairly large standard deviations to avoid excessive reliance on the prior distribution.

**Table 1. Values for a prior distribution ($x_a$ and $S_a$) and ranges.**

| Variable | Mean | Standard deviation | Range |
|---|---|---|---|
| COT | 10 | 64 | $0 - 128$ |
| CER [μm] | 10 | 30 | $2 - 60$ |
| ICOTF[c] | 0.5 | 1.0 | $0.0 - 1.0$ |
| CTH [km] | 2 | 10 | $0 - 20$ |
| CBH [km] | 1 | 10 | Max(LCL[a], Elev[b]) – CTH |

[a]LCL: Lifted Condensation Level; [b]Elev: Elevation; [c]ICOTF: ice COT fraction

### 2.2.2 Forward model

The TOA reflectance and radiance measured in the SGLI channels were simulated using radiative transfer calculations. In this study, we utilized an accurate radiative transfer code, RSTAR (Nakajima and Tanaka, 1986, 1988; Stamnes et al., 1988), which incorporates the scattering properties of cloud particles based on Mie scattering for liquid clouds and non-spherical Voronoi shapes for ice clouds (Ishimoto et al., 2012; Letu et al., 2012, 2016). The RSTAR version 7 (RSTAR7) package includes gas absorption line tables compiled into narrow bands using the k-distribution method. However, the k-distribution table is based on the HITRAN 2004 molecular spectroscopic database (Rothmana et al., 2005) and does not incorporate recent updates to the oxygen absorption lines.

Fast radiative transfer calculations employing various approximation techniques are commonly used in cloud retrieval (e.g., Nakajima and Nakajima, 1995; Walther and Heidinger, 2012; Hayashi, 2018). Our cloud retrieval algorithm similarly utilized an updated version of a fast forward model implemented by Nagao and Suzuki (2021) with the following two modifications: For VN9, we introduced cloud base pressure, which is more directly related to the amount of oxygen within clouds compared to CBH, as a new variable to account for

oxygen absorption within clouds. In the original version, CGT was assumed to be zero; For TI1 and TI2, radiative transfer computations in the TIR region were implemented based on techniques provided in previous studies (e.g., Nakajima and Nakajima, 1995; Kawamoto et al., 2001). The technical details of the forward model are provided in the supplemental material (see Text S1).

The forward model used in this study makes several assumptions regarding the cloud microphysical structure. The model assumed a plane-parallel cloud layer with a mixture of liquid and ice phases. The partitioning of the two phases is expressed in the ICOTF, and CER is assumed to be common between the liquid and ice particles. These assumptions, some of which are inconsistent with actual clouds that have vertically inhomogeneous profiles of cloud properties, including multi-layer clouds, are potential sources of retrieval bias.

## 2.3 Input data

This section outlines the input data used in the analyses presented in Sects 3 and 4. Our cloud retrieval algorithm was implemented using the following three SGLI standard products as inputs: The first input was the TOA radiance product (referred to as SGLI-LTOA), which provides SGLI-measured reflectances and radiances with an instantaneous field of view of 1 km, along with sun–satellite geometry conditions. The second input was the cloud flag product (referred to as SGLI-CLFG), which provides a confidence level indicating the presence or absence of clouds for each pixel (Nakajima et al., 2019). Additionally, this cloud flag product includes various flags related to surface conditions, such as snow/ice and sun glint. The third input was the SGLI land surface reflectance product (referred to as SGLI-RSRF). It provides land surface reflectance for the VNIR-to-SWIR channels of the SGLI, along with the parameters for the bidirectional reflectance distribution function model, which are input into the forward model for land pixels. For ocean pixels, the RSTAR7 subroutine was employed to estimate sea surface reflectivity from sun–satellite geometry and sea surface wind speed, which were then fed into the forward model. Notably, we effectively corrected for the impact of land and sea surface reflectance on the SGLI observed radiances in the manner described above. However, our algorithm did not explicitly account for the presence of sea ice over the ocean or its high reflectance. Therefore, we carefully excluded sea ice pixels by employing the sea ice fraction data provided by the Group for High Resolution Sea Surface Temperature (GHRSST) Level 4 datasets (Chin et al., 2017). In addition, several meteorological variables, such as the temperature profile, water vapor profile, surface pressure, surface temperature, and sea surface wind speed, were obtained from the Modern-Era Retrospective Analysis for Research and Applications Version 2 (MERRA-2) product produced by the U.S. NASA Global Modeling and Assimilation Office, as input to the forward model. Surface elevation data were obtained from the Terra Advanced Spaceborne Thermal Emission and Reflection Radiometer (ASTER) Global Digital Elevation Model (GDEM) Version 2 (ASTGTM v002) with a spatial resolution of approximately 30 m, developed jointly by NASA and Japan's Ministry of Economy, Trade, and Industry (METI). It should be noted that the uncertainties

in the meteorological variables provided by the MERRA-2 reanalysis data, as well as those in the land surface reflectance data from the SGLI-RSRF, were not explicitly accounted for in the inverse estimation. Furthermore, the impacts of aerosols on the observed radiance were not considered in either the forward model or the inverse estimation.

The cloud retrieval algorithm was applied to SGLI data for 2021 and 2022 to conduct regional and global analyses. SGLI offers products with different spatial resolutions. For the regional analysis, the SGLI tile product in the form of sinusoidal tiles, projected at a resolution of approximately 1 km (referred to as SGLI-LTOAK), was used. In contrast, for the global analysis, the SGLI global product resampled to a 1/24 degree resolution (referred to as SGLI-LTOAF) was used.

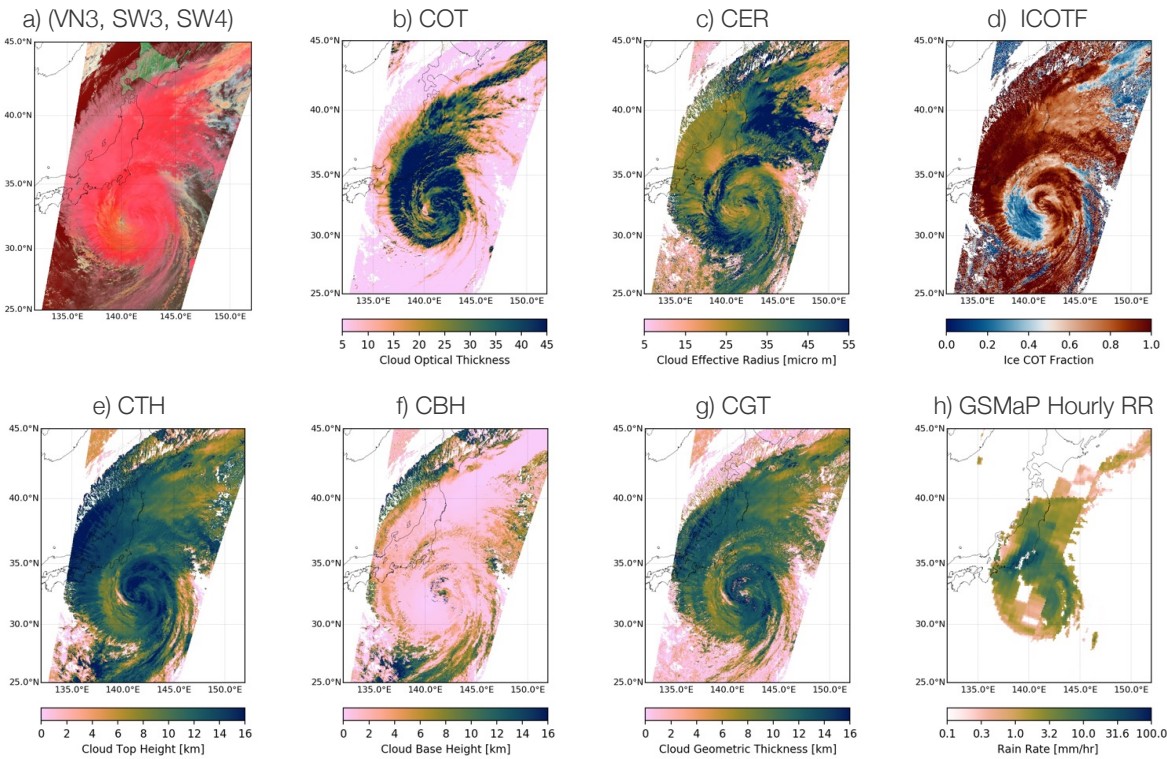

**Figure 3: (a)** Color composite image from SGLI VN3, SW3, and SW4 channels (centered at 443 nm, 1.63 μm, and 2.21 μm, respectively) for the area from 132–152°E and 45–25°N around Japan, observed at around 01:15 UTC (around 10:30 AM local sun time) on 1 October, 2021. **(b)** Similar to (a) but for COT retrieved via the cloud retrieval algorithm developed in this study. **(c)** Similar to (b) but for CER. **(d)** Same as (b) but for ICOTF. **(e)** Same as (b) but for CTH. **(f)** Same as (b) but for CBH. **(g)** Same as (b) but for CGT. **(h)** Hourly rain rate (RR) from the Japan Aerospace Exploration Agency (JAXA) Global Satellite Mapping of Precipitation (GSMaP) product.

### 3. Results

#### 3.1 Regional analysis

In this section, we present the results of applying our cloud retrieval algorithm to the SGLI measurements, focusing on a specific scene for demonstration and evaluation purposes. Figure 3a shows an RGB color composite image using SGLI VN3 (443 nm), SW3, and SW4 reflectances assigned to the red, green, and blue

channels, respectively. In general, ice clouds exhibit strong absorption in the SWIR region compared to that of liquid water clouds, resulting in ice clouds appearing deep pink and liquid clouds appearing light pink or white. The image was obtained on 1 October, 2021, for the area 132–152°E and 45–25°N around Japan, capturing a typhoon at its center. Typhoons are suitable targets for evaluating our cloud retrieval algorithm because they contain clouds of varying heights and thicknesses, including cumulonimbus, cumulus, and cirrus clouds.

However, the vertically inhomogeneous structure of typhoon clouds, including multi-layered structures, highlights potential limitations of the algorithm.

The horizontal distributions of the cloud properties retrieved from our algorithm were qualitatively consistent with the typhoon structure, as shown in Fig. 3b–3g. Figure 3b–3d shows the retrievals of the COT, CER, and ICOTF. The High COT values correspond to the dense cumulonimbus clouds associated with the typhoon's eyewall, whereas the low COT values surrounding them are consistent with the presence of cirrus clouds. Evaluating the CER is fraught with difficulty: While the high CER area corresponds to the rainfall area, as visualized by JAXA's Global Satellite Mapping of Precipitation (GSMaP) product (Fig. 3i), certain discrepancies were found, especially in regions north of the typhoon (around 145°E, 40°N) and in cirrus clouds over the Japan Sea and south of the typhoon. ICOTF provided a reasonable quantification of coloration in the RGB image shown in Fig. 3a, with high ICOTF values observed in the typhoon eyewall and cirrus cloud regions, indicating a dominance of the ice phase contribution. Conversely, low ICOTF values, indicating a dominance of the liquid phase, were observed in the cumulus region in the upper right of the image and at the eye of the typhoon. The moderate ICOTF values around 145°E and 40°N may have been influenced by multi-layer clouds. These results for the COT, CER, and ICOTF retrievals are consistent with those obtained using the cloud phase retrieval algorithm proposed by Nagao and Suzuki (2021), which is one of the sources of our algorithm. In other words, the incorporation of the TIR and oxygen A-band channels in this study did not adversely impact the quality of cloud phase retrieval based on the SWIR channels.

Figure 3e–3h presents the retrievals of CTH, CBH, and CGT, respectively. These primarily rely on information from the TIR and oxygen A-band channels. The horizontal distributions of these retrievals were qualitatively consistent with the three-dimensional structure of the typhoon. However, certain biases in CTH were observed in areas with multi-layer clouds, including cirrus clouds overlying low clouds. For instance, low CTH values were observed in the upper right of the image. This bias can be attributed to the fact that the proposed algorithm employs a single-layer cloud model. Naturally, CTH biases lead to CGT biases. It remains unclear how to interpret the CTH, CGT, and CBH retrieved from multi-layer cloud pixels in terms of cloud microphysics. Although such biases were expected owing to the limitations of our algorithm, the overall retrievals of cloud vertical structure using the TIR and oxygen A-bands performed well.

## 3.2 Global analysis

Our cloud retrieval algorithm was subsequently applied to the SGLI global products, which were resampled to a 1/24 degree resolution, as described in Sect 2.3. We used 48 files for the 1st, 9th, 17th, and 25th of each month from January to December, 2021. Figure 4a–4f shows the global geographical distributions of the mean

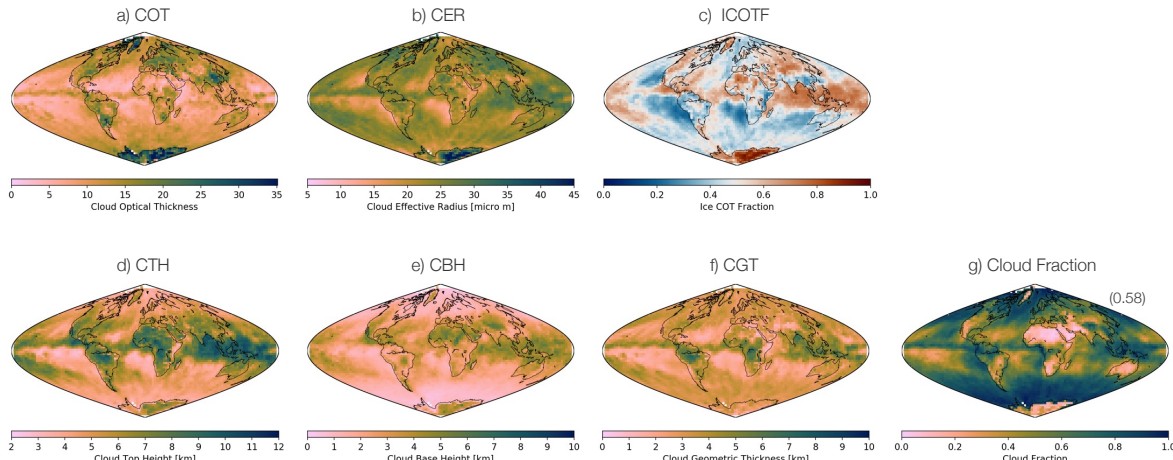

**Figure 4: (a) Global geographical distribution of the annual mean COT for 2021, derived from the SGLI global radiance product using the cloud retrieval algorithm developed in this study. (b) Similar to (a) but for CER. (c) Same as (a) but for ICOTF. (d) Same as (b) but for CTH. (e) Same as (a) but for CBH. (f) Same as (a) but for CGT. (g) Same as (a) but for cloud fraction, with the number in parentheses indicating the global mean of the cloud fraction.**

values for the COT, CER, ICOTF, CTH, CBH, and CGT retrievals. Figure 4g shows the cloud fraction, which represents the number of cloud pixels detected by the SGLI standard cloud flag product and subsequently processed by our cloud retrieval algorithm.

The distributions of COT, CER, and ICOTF in Fig. 4a–4c indicate that our comprehensive retrieval algorithm performed stably for global observations. First, no artificial gaps were found in the distribution over the ocean and land areas. In addition, there was no noticeable impact of high surface reflectivity, such as sun glint and snow/ice cover, except in Greenland, the Arctic, and Antarctica. In these specific polar regions, retrievals of COT, CER, and ICOTF exhibited unrealistically high values. These anomalies were associated

with the misidentification of snow/ice surfaces as clouds in clear pixels or the high reflection of snow/ice surfaces under optically thin clouds. The distributions of COT, CER, and ICOTF in Fig. 4a–4c were quantitatively consistent with those retrieved using a combination of SW1, SW3, and SW4, shown by Nagao and Suzuki (2021). This result emphasizes that the modification from Nagao and Suzuki (2021), where CTH and CBH were also retrieved simultaneously by introducing a 763 nm channel (VN9) and TIR channels (TI1

and TI2), does not adversely impact the retrievals of COT, CER, and ICOTF.

    The global distribution of CTH in Fig. 4d was qualitatively consistent with the well-known characteristics of various cloud regimes. High CTHs are observed in regions characterized by deep convective clouds, particularly in the Intertropical Convergence Zone (ITCZ), which forms near the equator, and the South Pacific Convergence Zone (SPCZ), which branches off from the ITCZ. In addition, high CTHs, which primarily

originated from cirrus clouds, are distributed over the Sahara. Low clouds, primarily in the form of extensive stratocumulus decks, are observed over the subtropical ocean off the western coasts of continents (e.g., off California, Peru, and Namibia). In contrast, another well-known low-cloud region in the southern Indian Ocean

is not clearly identifiable in Fig. 4d but can be identified as a low ICOTF in Fig. 4c, indicating that liquid water clouds are dominant.

The global distribution of CBH in Fig. 4e exhibits both similarities and differences when compared to the CTH distribution in Fig. 4c, both of which support the functionality of our algorithm for estimating CGT. First, several cloud regimes described in the preceding paragraph (i.e., deep convective clouds in the ITCZ and SPCZ, cirrus clouds in the Sahara, and stratocumulus decks off the west coasts of the continents and the southern Indian Ocean) can also be identified in Fig. 4e. Conversely, notable differences exist between the spatial patterns

of CTH and CBH over mid- and high-latitude oceanic regions. In particular, the low-CTH regions originating from the stratocumulus decks off Peru and Namibia appear to be significantly lower and more isolated than their surroundings, whereas the CBH of these stratocumulus decks lacks a clear boundary with the CBH over the Southern Ocean. A similar difference was observed between another stratocumulus deck off California and the clouds over the Bering Sea and Arctic Ocean. The validity of these distinctions was supported by comparing

the CTH and CBH products derived from the satellite-based active sensors, as discussed in Sect 4 below. These results qualitatively support the idea that our algorithm effectively separates the CTH and CBH information co-located in the TOA reflectance in the 763 nm channel.

The global distribution of CGT in Fig. 4f also agrees with typical climatological cloud regimes and was similar to that of CTH. For example, mean CGTs of less than 2 km were observed in cumulus cloud regions

within the trade-wind zone and in stratocumulus decks over the ocean off the western coasts of continents, whereas mean CGTs greater than 6 km corresponded to deep convective clouds in the tropics. In addition, correlations between the spatial distribution of CGT and other cloud properties, such as COT, CER, and ICOTF, also appear to be associated with typical cloud regimes.

However, Fig. 4f also raises concerns regarding retrieval biases depending on the cloud type. The first

concern is the overestimation of CGT for cirrus clouds over the Sahara. Cirrus clouds are generally characterized by a thin COT, thin CGT, and high CTH. Comparing Figs. 4a, 4d, and 4f, the relatively low COT and high CTH in the Sahara compared to its surroundings are consistent with the characteristics of cirrus clouds. However, the coincident CGT, at approximately 3–4 km, is not very thin, which seems inconsistent with the spatial pattern of COT. Conversely, the second concern is the underestimation of CGT for deep convective

clouds in the ITCZ and SPCZ regions. While Fig. 4d illustrates very high CTHs in these regions, the difference in CGT between the areas inside the ITCZ and SPCZ and those outside was not as pronounced as that in CTH. The third concern is the overestimation of CGT for stratocumulus decks off Peru, where the CGT is relatively large compared to its surroundings. If there are doubts regarding Fig. 4d, the coincident CTH off Peru also appears to be higher than that of its surroundings. These concerns regarding the biases in CGT retrieval,

naturally related to biases in CTH and CBH retrievals, are addressed in Sect 4, where comparisons with similar statistics from satellite-borne active sensors are performed.

## 4. Validation

This section presents a quantitative validation of the CBH retrievals obtained by applying our algorithm to
actual SGLI measurements. Validation was conducted by comparing the SGLI-derived CBH with the CBHs
measured using ground-based and ship-borne ceilometers. Furthermore, a global statistical analysis was
conducted to comprehensively compare the SGLI-derived CTH and CBH with those obtained from CloudSat
and CALIOP observations. This will enable us to characterize the systematic bias of not only the SGLI-derived
CTH and CBH, but also their differences in CGT.


### 4.1 Potential uncertainty in CBH retrieval

We should prepare for a certain level of uncertainty in the CBH obtained using our algorithm from SGLI
measurements. A preliminary estimation of the potential uncertainty and its characteristics is needed before
comparing the CBHs derived from SGLI and ceilometers to facilitate the interpretation of the validation results.
First, the CBH retrieval, obtained through the combination of the oxygen A-band and TIR channels, should
have at least an equivalent degree of potential uncertainty as the CTH retrieval. As illustrated in Fig. 2, the TOA
reflectance from the SGLI VN9 exhibited comparable sensitivity to both CBH and CTH. This implies that the
uncertainty in CTH retrieval propagates to CBH retrieval through changes in VN9's TOA reflectance. In
essence, the upper limit of the accuracy of CBH retrievals with VN9 is constrained by the accuracy of the CTH
retrievals. In addition, the uncertainty of the CTH retrieved using TIR channels depends on the COT and cloud
vertical structure. For example, the CTH retrieval provided in the MODIS cloud property product (MOD06),
which is derived using a combination of MODIS TIR channels, has an uncertainty within 1 km for mid-level
water clouds, based on comparisons with CALIOP. In contrast, for thin cirrus clouds, CTHs are underestimated
by 2–3 km relative to those detected by CALIOP due to factors such as COT and overlap with lower-level
clouds (Baum et al., 2012; King et al., 2013). In light of these considerations, it would be reasonable to assume
that the CTH retrieval using the SGLI TIR channels may exhibit uncertainties comparable to those of the
MOD06 product.

Given the potential uncertainty in TIR-derived CTH retrieval and its subsequent impact on CBH retrieval, it
would be prudent to predict uncertainties of approximately 1 km for water clouds and 2–3 km for ice clouds
when employing our algorithm for CBH retrieval. The potential accuracy of CBH retrieval based on SGLI VN9
measurements is likely lower than the vertical resolution achievable with active sensors on satellites.

In our algorithm, the uncertainty in CBH retrieval is also entangled with the uncertainty in COT retrieval.
We performed a sensitivity analysis based on the error propagation theory to examine how measurement
uncertainties propagate to retrieval uncertainties (see Text S2 in the supplemental material). Figure S1
demonstrates how perturbations in individual measurement channels induce retrieval errors. Notably,

perturbations in SW1, which is a channel sensitive to COT but located outside the oxygen A-band, can induce errors not only in COT retrieval (Fig. S1(1,1)) but also in CBH retrieval (Fig. S1(5,1)). This indicates that COT errors disturb the separation of COT and CBH from VN9 measurements. Figure S2 further demonstrates how the overall uncertainty in the multi-wavelength measurements incorporated into the inverse estimation propagates to retrieval uncertainties. The comparison of Figs. S2a1 and S2b1 reveals incorporating VN11 alongside SW1 reduce the uncertainty in COT retrieval, which, in turn, contributes to reduce uncertainty in CBH retrieval. As described in Section 2.2.1, our algorithm utilized both SW1 and VN11. The results of these sensitivity analyses emphasize the importance of carefully addressing uncertainties in COT retrieval when deriving CBH from VN9 measurements. The entanglement of COT, CTH, and CBH retrieval errors associated with oxygen A-band measurements has also been reported by Lelli et al. (2014).



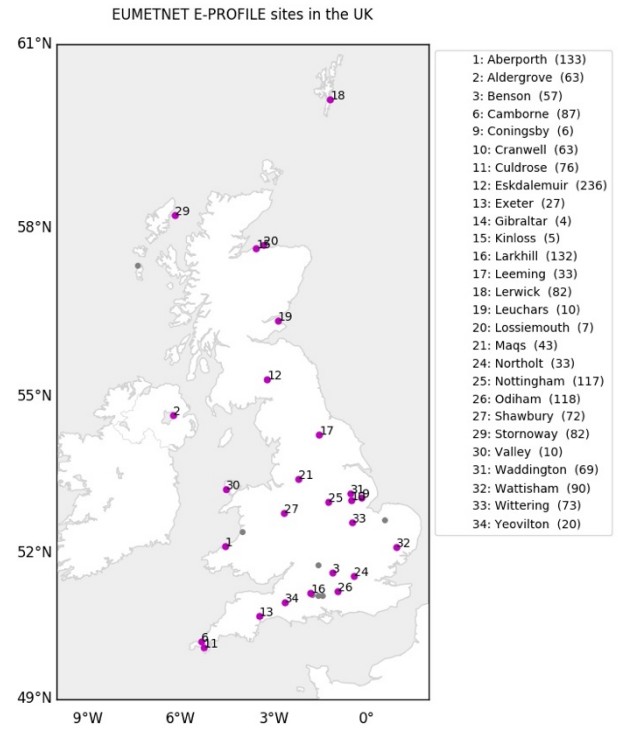

**Figure 5: Distribution of EUMETNET E-PROFILE sites across the United Kingdom. Magenta points indicate sites used in this study, whereas gray points represent sites excluded from the analysis. The values in parentheses in the legend indicate the elevation of each site in meters.**

### 4.2 Comparison with ground-based ceilometer measurements

This subsection presents a comparison of the CBH data derived from SGLI measurements with the CBH data from ground-based ceilometers. We used ceilometer data produced by the EUMETNET E-PROFILE as validation data, which were obtained from the Centre for Environmental Data Analysis archive. The EUMETNET E-PROFILE sites have been deployed in various European countries. In this study, data from sites in the United Kingdom were used, as shown in Fig. 5. The ceilometer data period spanned from September 2021 to December 2022, with varying levels of data availability across different sites. The SGLI CBH data were retrieved based on the methodologies and input data described in Sect 2, utilizing the SGLI radiance product projected at a resolution of 1 km.

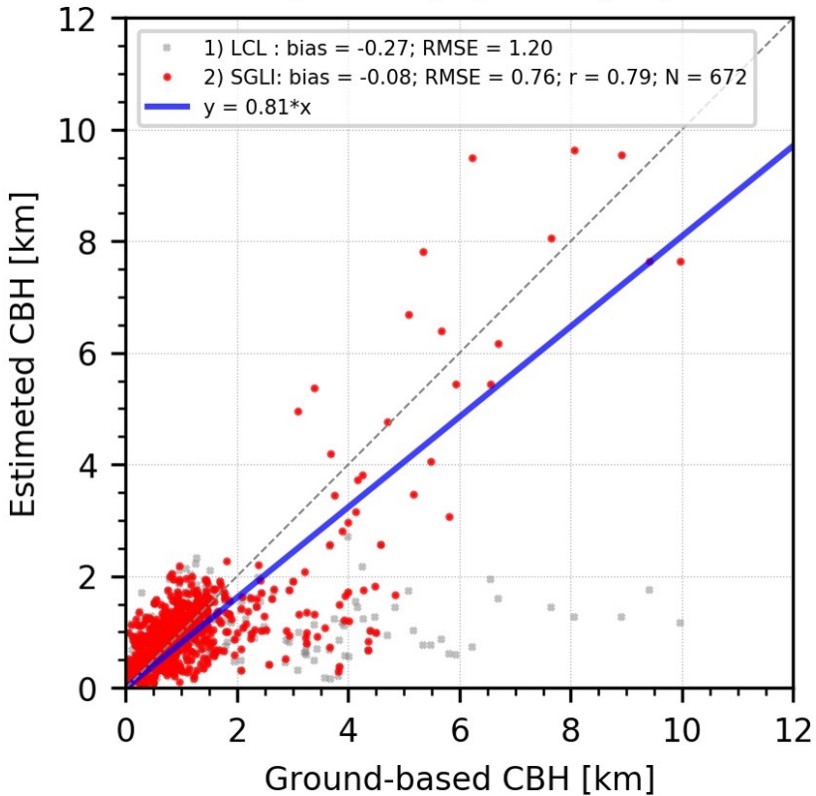

**Figure 6: Comparison of CBHs retrieved from SGLI measurements with CBHs measured using ground-based ceilometers at EUMETNET E-PROFILE sites across the United Kingdom. Red points represent CBH retrievals from SGLI, while gray points denote LCLs estimated from MERRA-2 products.**

A comparison of the CBH data derived from the SGLI and ground-based ceilometers is presented in Fig. 6 (red dots). The CBH data plotted in Fig. 6 represent average values rather than instantaneous values. Averaging was applied to reduce variability due to discrepancies in the fields of view and observation times between the SGLI and ceilometers. Specifically, the SGLI CBHs were spatially averaged within a 4 km radius of the sites, while the ground-based CBHs were temporally averaged over a 30 min period before or after the SGLI observation time. In addition, only ground-measured CBH data with standard deviations (SD) smaller than 1 km within this temporal range were used for comparison. This is because data with significant temporal variability are more likely to encompass diverse cloud types, such as cumulus and cirrus clouds. Notably, these three thresholds ($\Delta s < 4$ km, $\Delta t < 30$ min, and $\text{SD}[CBH_{ground-based}] < 1$ km) were empirically determined. However, it is important to note that the data screening using this $\text{SD}[CBH_{ground-based}] < 1$ km might unintentionally exclude scenes and cloud types that are challenging to retrieve. This is because factors that cause significant biases in cloud retrievals, such as sub-pixel scale heterogeneity of COT and CBH, are more

likely to be common in scenes with diverse cloud types, such as cumulus and cirrus clouds, which are filtered
out by this screening. Therefore, it should be noted that the SGLI cloud properties retrieved using our algorithm
may contain lower-quality data than those presented in the validation results here.

    Figure 6 illustrates that the SGLI CBHs exhibited a high degree of agreement with the ground-based CBH, as evidenced by a high correlation coefficient ($r = 0.79$), a significantly small bias of −80 m, and a moderate RMSE (~790 m). However, closer examination of the plot reveals that the data can be divided into three distinct
groups, each with different error characteristics. The first group comprises data with ground-based CBHs lower than 2 km.  This group was distinguished by satisfactory concordance between the CBHs derived from the SGLI and the ground-based ceilometer, with a slight bias of approximately 30 m and an RMSE of approximately 480 m. The second group comprised data with ground-based CBH values ranging from 2 to 4 km. In this group, the SGLI-derived CBHs consistently ranged between 0 and 2 km, exhibiting a significant negative bias. The third
group comprised data with ground-based CBH values exceeding 4 km. The SGLI-derived CBHs of this group exhibited a high degree of agreement with the ground-based CBH values, as evidenced by a correlation coefficient of approximately 0.72. The bias and RMSE of the CBH of this group were approximately −250 m and 1.5 km, respectively. Notably, the RMSEs for the first and third groups (480 m and 1.5 km, respectively) were not only consistent with the potential uncertainties for water and ice clouds (1 km and 2–3 km,
respectively) estimated in the preceding section, but were also relatively smaller. This result confirms that the proposed algorithm performs reasonably well within the retrieval framework described in Sect 2.

    It should be emphasized that the thresholds ($\Delta s$, $\Delta t$) can influence the results in Fig. 6, but are not critical. Figure S3 in the supplementary material illustrates the dependence of bias, RMSE, and correlation coefficient between CBHs from SGLI and ceilometers on ($\Delta s$, $\Delta t$). It demonstrates that bias and RMSE worsen when $\Delta t$
is set to shorter than 30 min. Additionally, it indicates that the choice of $\Delta s < 4$ km and $\Delta t < 30$ min is not the only method to minimize bias and error. In other words, there are alternative values for ($\Delta s$, $\Delta t$) that can yield better agreement in CBH between SGLI and ceilometer.

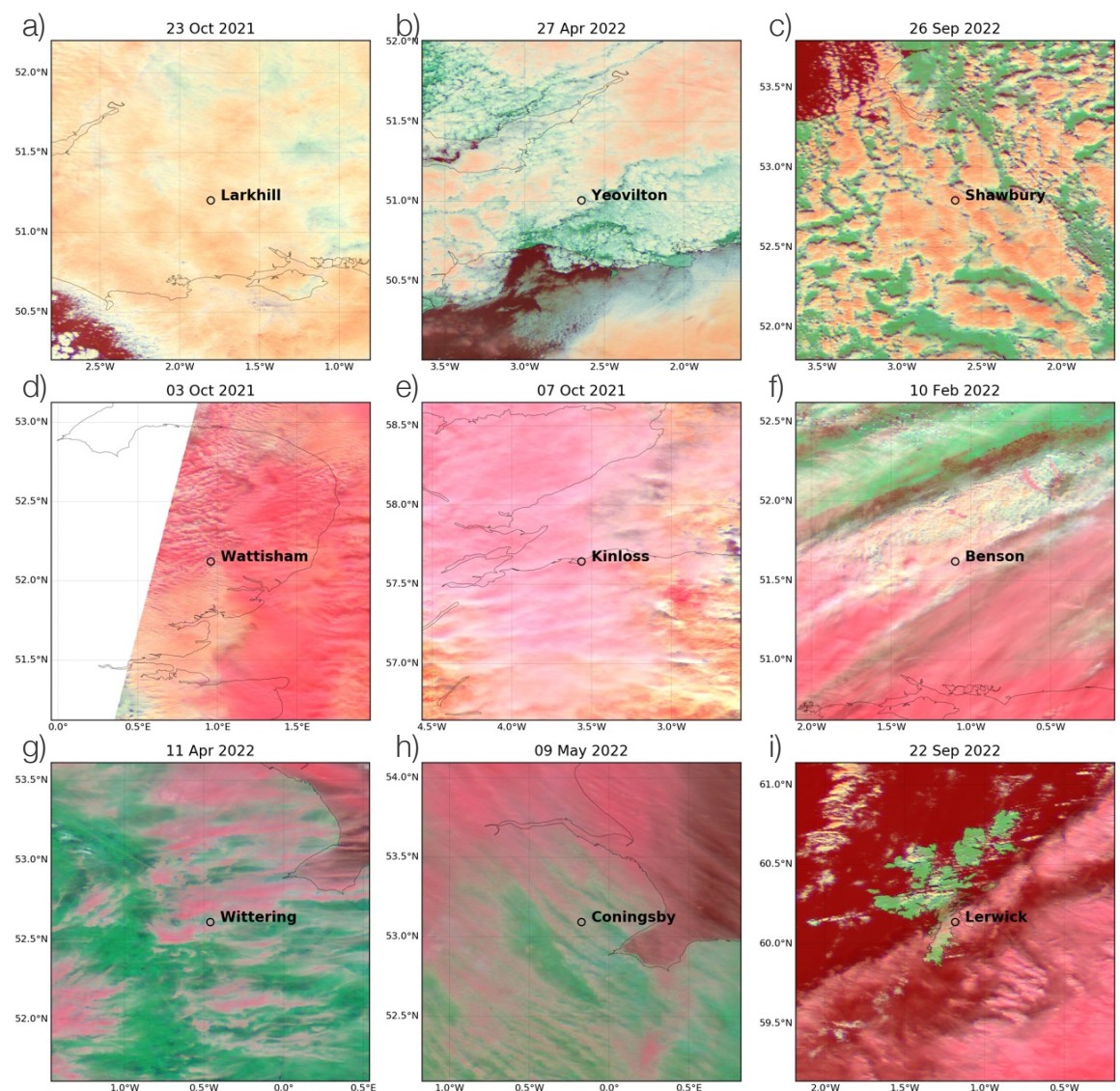

**Figure 7: Color composite images from the SGLI VN3, SW3, and SW4 channels (centered at 443 nm, 1.63 μm, and 2.21 μm, respectively). These images were selected from data with ground-measured CBHs of: (a–c) 2 km or lower, (d–f) 2–4 km or lower, and (g–i) 4 km or higher, ensuring that the sites and dates do not overlap.**

regime. The top, middle, and bottom images in Fig. 7 were selected from the first, second, and third groups, respectively. The images in Fig. 7 are RGB color composites in which the SGLI VN3, SW3, and SW4 channels (centered at 443 nm, 1.63 μm, and 2.21 μm, respectively) are assigned to red, green, and blue, respectively. The color tone (brightness and hue) of these color composites is indicative of cloud properties. First, the brightness of the image increased with higher COT values, as all three channels are sensitive to COT. In contrast, the hue indicates the cloud thermodynamic phase and CER, to which SW3 and/or SW4 are sensitive. In particular, the

reflectance at SW3 was relatively large for water clouds and small for ice clouds owing to absorption by ice
particles; however, in comparison, the reflectance at SW4 did not vary significantly for different cloud phases. Conversely, both reflectances at SW3 and SW4 exhibited a reduction with an increase in CER. These characteristics of the three channels resulted in water clouds appearing white to orange (Fig. 7a–c), while ice clouds exhibited a reddish hue (Fig. 7g–i). The multi-layered structure of ice clouds overlying water clouds is often identified by the horizontal distribution of the brightness temperature. Nevertheless, the color composite
in Fig. 7 can also be utilized to identify this structure through the overlap between the whitish representation of water clouds and the reddish representation of ice clouds (Fig. 7d–f).

The first group was characterized by extensive homogeneous liquid-phase low-level clouds, such as stratocumulus and cumulus clouds (Fig. 7a–c). Figure 7d–f illustrates representative images of the second group, which exhibited multi-layer cloud structures with optically thin middle and upper clouds overlying the lower
clouds. Although it is challenging to quantitatively predict how our retrieval algorithm (which assumes a single-layer plane-parallel cloud structure) will behave for such multi-layer clouds, it is evident that our algorithm tends to output CBH retrievals that are close to the CBH of lower clouds in such multi-layer cloud cases. Although not all scenes in this group exhibited a multi-layer structure, they generally exhibited more pronounced horizontal heterogeneity in cloud properties than the first group (Fig. 7a–c). The third group was
dominated by mid- and high-level clouds, as suggested by the relatively high CBH (> 4 km), and the examples shown in Fig. 7g–i indicate that they are not as optically thick as the first and second groups. Moreover, the reddish hue of the RGB composite indicated the presence of ice within the clouds. In summary, the first group consisted of homogeneous single-layer liquid-phase low clouds, the second group consisted of multi-layer clouds, and the third group consisted of relatively thin, ice-containing mid- and high-level clouds.

The CBH retrievals using the SGLI oxygen A-band were most effective for mid- and high-level clouds. The gray points in Fig. 6 represent the LCL, which was estimated using MERRA-2 reanalysis data. Although our algorithm does not use the LCL as an initial value, as shown in Table 1, the difference between red and gray corresponds to the update of information obtained by retrieving the CBH with the SGLI VN9 relative to the LCL value. The largest update was observed in the third group. Conversely, in the first group, the distribution
of the CBH retrievals, indicated by the red points, overlapped with the distribution of the LCL indicated by the gray points. This suggests that for clouds with such low CBH values, the SGLI CBH retrievals may not significantly enhance the CBH information.

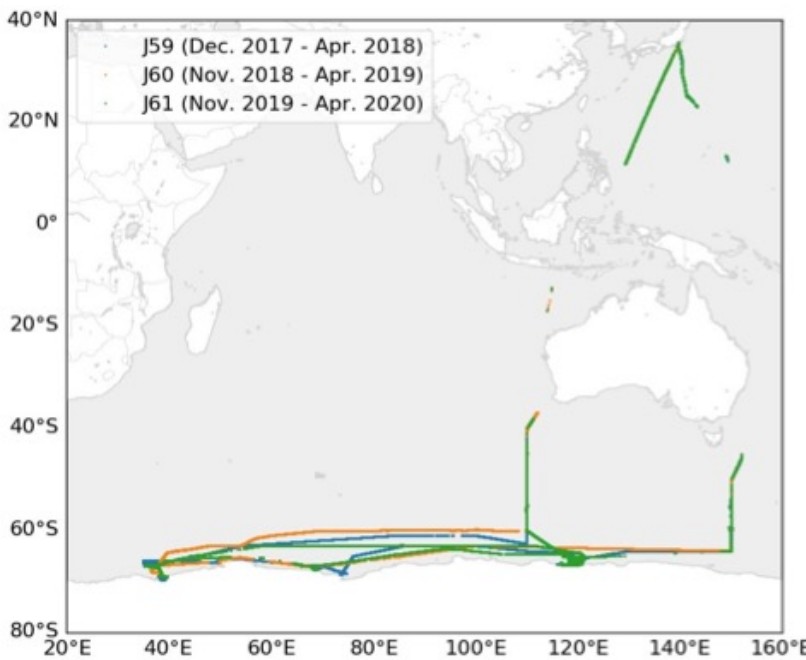

**Figure 8: Observations of CBH conducted using a shipborne ceilometer installed on the Antarctic research vessel Shirase during the Japanese Antarctic Research Expeditions 59, 60, and 61. The observations took place from December 2017 to April 2018, November 2018 to April 2019, and November 2019 to April 2020, respectively. Note that these observations do not overlap with the exclusive economic zones of any countries other than Japan.**

## 4.3 Comparison with ship-borne ceilometer measurements

This section presents a comparison of the SGLI-derived CBH data with CBH data measured using a ship-borne ceilometer installed on the Antarctic research vessel *Shirase*. The ceilometer-based CBH observations were conducted during research cruises as part of the Japanese Antarctic Research Expeditions 59, 60, and 61 (Hirasawa et al, 2022). The cruises were conducted from December 2017 to April 2018, November 2018 to April 2019, and November 2019 to April 2020, respectively. The voyages departed from Japan, proceeded southward, and subsequently returned after touring the Southern Ocean. Figure 8 shows the observation points for the ceilometer-based CBH data. It should be noted that this study employed CBH data recorded in the Southern Ocean (south of 40°S) and south of Japan (10°N–35°N) to avoid the exclusive economic zone of any country other than Japan.

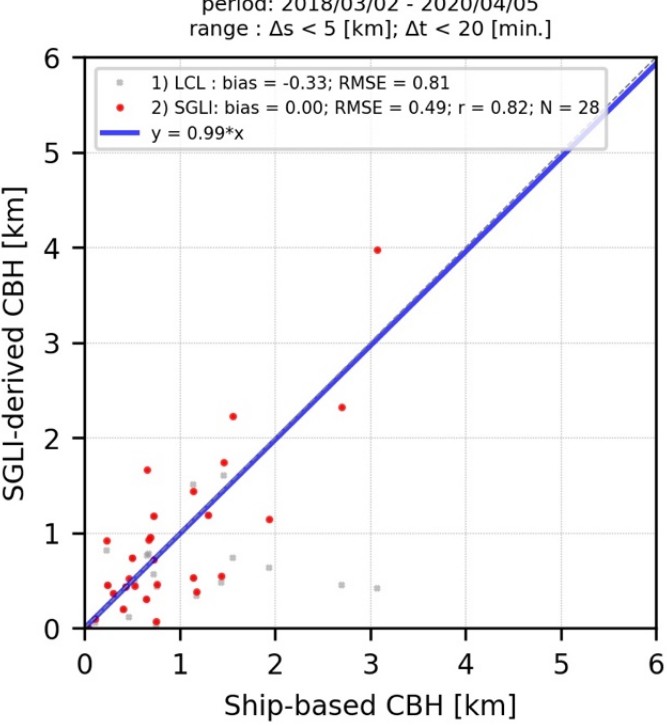

**Figure 9: Comparison of CBHs retrieved from SGLI measurements with those measured using ship-borne ceilometers on the Antarctic research vessel Shirase. Red points represent CBH retrievals from the SGLI, while gray points indicate LCLs estimated from MERRA-2 products.**

Figure 9 presents a comparison between the CBHs derived from the SGLI and those measured by the ship-borne ceilometer, with the data represented by red points. Similarly to Fig. 6, Fig. 9 presents the averaged values obtained using the following procedure. The SGLI-derived CBHs were spatially averaged within a 5 km radius of the Shirase location, while the ceilometer CBHs were temporally averaged over a 20 min period before and after the SGLI observation time. Data from the regions of the ocean covered by sea ice, which significantly

impacts cloud property retrieval, were strictly excluded as follows: First, all data at latitudes south of 66.6°S were excluded due to the severe degradation of the SGLI standard cloud flag product quality in these regions (Nakajima et al., 2019), which tends to misidentify "clear" pixels covered by sea ice as "cloudy" pixels. This misidentification is due to the cloud detection test and threshold in the SGLI cloud flagging algorithm switching at a latitude of 66.6°. In addition, specific dates and times when sea ice was observed along Shirase's route were

excluded based on the voyage records of the Shirase (Kuji et al., 2018). The aforementioned rigorous screening process yielded a limited number of data points that were utilized for comparison, as depicted in Fig. 9. Consequently, data with a ceilometer CBH > 4 km could not be obtained for analysis.

    Figure 9 shows that the SGLI-derived CBH exhibited a high degree of agreement with the ship-borne ceilometer CBH. This was evidenced by a high correlation coefficient of $r = 0.82$, a bias of 0 m, and an RMSE

of 490 m, which were smaller than the bias RMSE when LCL was used as the CBH estimate (−330 m and 810 m, respectively). The behavior of the SGLI CTH and LCL relative to the ceilometer CBH demonstrated a notable change around the ceilometer CBH of approximately 1.5 km. For a CBH of < 1.5 km, both the SGLI CBH and LCL were in good agreement with the ceilometer CBH. However, for a CBH of > 1.5 km, the SGLI CBH provided more accurate estimates for the ceilometer CBH than LCL. Fig. 6 of the preceding section

illustrated that a negative bias was identified in the SGLI CBH for data with a ceilometer CBH of 2–4 km. This was indicative of a multi-layer cloud structure, as shown in Fig. 7d–f. However, no analogous bias was identified in Fig. 9. This discrepancy is not attributable to the disparate behavior of our algorithm; rather, it is simply because such multi-layer clouds are not included in the dataset shown in Fig. 9.

**4.4 Statistical comparison of CTH, CBH, and CGT with CloudSat/CALIPSO observations**

This section presents a comprehensive validation of both CTH and CBH derived from SGLI, complementing the validation of CBH alone presented in the previous two sections. As illustrated in Fig. 2, the accuracy of CBH retrieval using the 763 nm channel is shown to depend on the accuracy of CTH retrieval. Therefore, in addition to the CBH-only validation, a thorough validation of both CTH and CBH retrievals is necessary.

Satellite-borne CloudSat/CPR and CALIPSO/CALIOP observations provide reliable sources for the comprehensive validation of both SGLI-derived CTH and CBH on a global scale. However, due to the entirely disparate orbits of GCOM-C (which follows a morning orbit), CloudSat, and CALIPSO (which follows an afternoon orbit), the region of overlap between the SGLI and CPR-CALIOP orbits (for which match-up data can be produced) is almost exclusively limited to polar regions and high latitudes. Consequently, in this study,

rather than creating a matched dataset of instantaneous CTH and CBH values derived from the SGLI and CPR-CALIOP observations, we prepared and compared zonal means of SGLI- and CPR-CALIOP-derived CTH and CBH values.

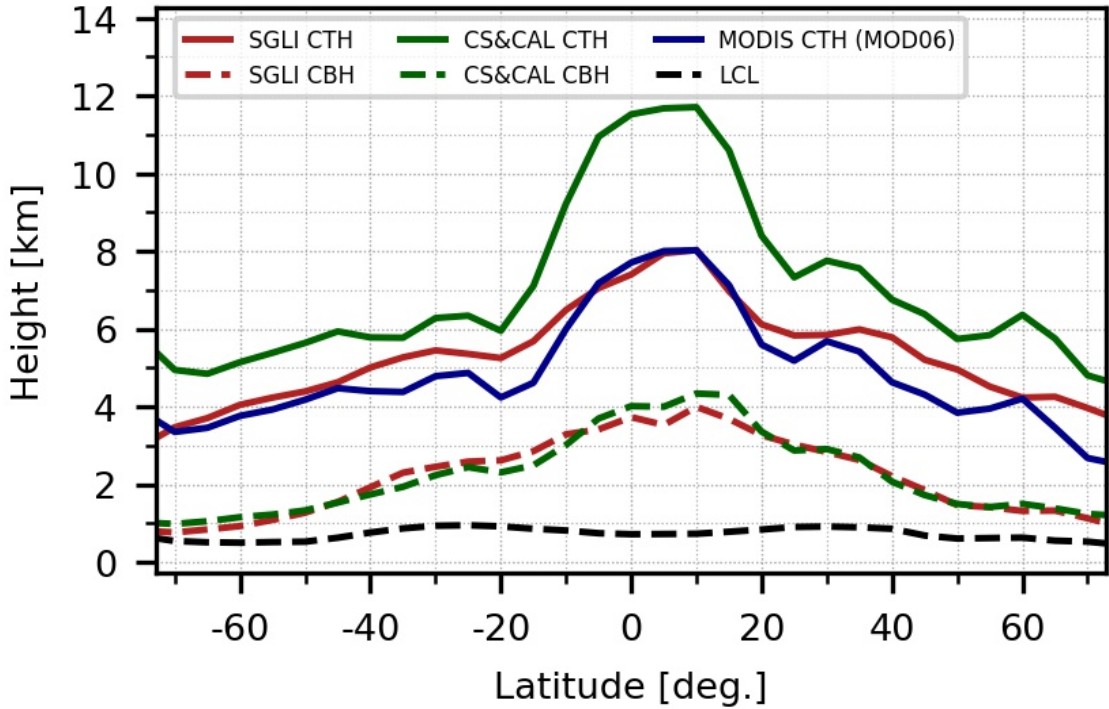

**Figure 10: Annual zonal means of CTH, CBH, and LCL over the global ocean. CTH is depicted by solid lines, while the CBH and LCL are indicated by dashed lines. The data were obtained from four different sources, each represented by a different color: (Red) CTH and CBH retrieved from the SGLI measurements, which are identical to those employed in Fig. 4; (Green) CTH and CBH identified using 2B-CLDCLASS-LIDAR R05 product, which combines CloudSat CPR and CALIPSO lidar observations; (Blue) CTH provided by the MOD06-1KM-AUX R05 product; and (Black) LCL estimated using MERRA-2 products.**

Figure 10 presents a comparison of the annual and zonal means of CTH and CBH derived from SGLI, CPR-CALIOP, and MODIS observations. The red lines represent the zonal means of the SGLI CTH (solid line) and CBH (dashed line), which are the same data used in Fig. 4. The green lines represent the zonal means of CTH (solid line) and CBH (dashed line) obtained from the 2B-CLDCLSS-LIDAR R05 product, which combines the CPR and CALIOP measurements to identify the vertical profile of clouds. The blue solid line represents the MODIS TIR-based CTH provided by the MOD06-1KM-AUX R05 product. This product contains a subset of the MODIS Cloud Properties Product (MOD06) Collection 6 (Platnick et al., 2017), which is collocated within each CPR footprint. The black dashed line represents the LCL estimated from the MERRA-2 reanalysis data.

The SGLI CBH was consistent with the CPR-CALIOP CBH, whereas the SGLI CTH exhibited a significant negative bias relative to CPR-CALIOP CTH. As demonstrated by the overlapping red and green dashed lines in Fig. 10, the zonal means of the SGLI CBH exhibited a high degree of agreement with those of the CPR-CALIOP CBH, with deviations within approximately 500 m across all latitudes. However, the zonal mean of

the SGLI CTH exhibited a significant negative bias of at least 2 km compared with that of the CPR-CALIOP CTH across all latitudes, reaching approximately 4 km in the tropics. Notably, an underestimation of at least 2 km in the SGLI CTH relative to the CPR-CALIOP results leads to a corresponding underestimation of at least 2 km in the CGT estimation, which is derived from the difference between CTH and CBH.

As discussed in Sect 4.1, previous studies have identified a similar bias in MOD06 CTH in comparison to
the CALIOP-detected CTH. This is attributed to the optically thin cirrus clouds and associated multi-layer clouds. Therefore, it is reasonable to assume that the SGLI CTH bias is caused by similar factors. Indeed, the SGLI CTH was generally consistent with the MOD06 CTH, as evidenced by the overlap of the red and blue solid lines.

The scenario in which an optically thin cloud layer near the cloud top induces a negative bias in the SGLI
CTH appears to be consistent with the result that a significant negative bias is not identified in the SGLI CBH but in the CTH. Given that the uncertainty in CTH is propagated to the uncertainty in CBH in searches using the 763 nm channel, it is remarkable that a significant negative bias is observed only for CTH.

There are two potential explanations for this significant finding. One is the presence of optically thin cirrus clouds overlying opaque clouds, which can only be detected by CALIOP but are transmissive to TIR radiation.
This arises because TIR typically retrieves cirrus clouds with COTs of 0.1 or greater (e.g., Iwabuchi et al., 2016), while the CALIOP lidar is capable of detecting optically thinner cloud layers with optical thicknesses as small as about 0.01 (e.g., Winker et al., 2009). In such vertical cloud structures, the CTH and CBH retrieved by our algorithm are expected to correspond to those of the opaque clouds. Conversely, the combined CPR-CALIOP observations would detect the CTH of the cirrus clouds and the CBH of the opaque clouds. Consequently, this
scenario could result in a negative bias with regard to the SGLI-derived CTH but not the SGLI-derived CBH.

Another possibility is error compensation between positive and negative biases of the SGLI-derived CBH depending on the cloud type, with their zonal sum close to zero. As a cause for negative bias, it was demonstrated that multi-layer cloud structures, such as the second group in Fig. 6 corresponding to Fig. 7d–f, can underestimate the CBH. Conversely, cloud structures that can induce a positive bias in SGLI CBH retrieval
should also be possible. One possible case, for example, is clouds with a precipitation layer near the cloud base, where large drops are sharply detected/resolved by CPR as strong echoes but might appear blurred in the 763 nm channels due to the small optical thickness in the visible region. This difference in sensitivity between CPR and 763 nm to the precipitating cloud base could induce a positive bias in SGLI CBH retrieval (i.e., SGLI CBH is higher than CPR CBH). If the negative and positive biases in SGLI CBH cancel each other zonally, the zonal
mean net bias might appear to be relatively insignificant.

Both of these potential explanations are related to the vertical inhomogeneity of the cloud property profile, including the multi-layer cloud structure. However, the inability of SGLI passive remote sensing to resolve the vertical profile makes it challenging to determine which of these two explanations contributes more to the results in Fig. 10.

It is also noteworthy that the zonal mean CBH derived from SGLI was in agreement with that retrieved from hyperspectral measurements in the oxygen A-band by SCIAMACHY. Figure S4 illustrates the seasonal variations of the SGLI-derived zonal mean CBH and the difference between the JJA and DJF months, which generally aligns with those previously reported by Lelli and Vountas (2018) for SCIAMACHY. However, the SGLI-derived zonal mean CBH was approximately 1 km higher than that from SCIAMACHY in the low-

latitude zone, including the peak value. Further investigation into the factors contributing to these quantitative differences between SGLI-derived and SCIAMACHY-derived CBHs, such as differences in algorithms, channels used, or cloud type dependencies, remain an important task for the future research.

**5 Summary and discussion**

**5.1 Conclusion**

This study describes a method for the retrieval of CBH and CGT based on the use of 763 nm channels of the GCOM-C/SGLI. The 763 nm channel, located within the oxygen A-band, can provide CBH and CGT through satellite-based passive remote sensing. Moreover, the challenge in utilizing the 763 nm channel is that it is

sensitive not only to CBH and CGT but also to CTH and other cloud properties. Therefore, this study implemented an algorithm by coupling two algorithms: the four-channel algorithm, which has been demonstrated with ADEOS-II/GLI, the predecessor of SGLI, for only water clouds, to retrieve COT, CER, CTH, and CGT using the VNIR, SWIR, 763 nm, and TIR channels (Kuji and Nakajima, 2001, 2004; Kuji 2013); and the cloud phase retrieval algorithm using multiple SWIR channels of SGLI (Nagao and Suzuki,

2021). As a consequence of this coupling, the algorithm can be seamlessly applied not only to water clouds but also to global clouds, including mixed-phase and ice clouds, and can simultaneously retrieve COT, CER, ICOTF (as the cloud thermodynamic phase), CTH, CBH, and CGT (as the difference between CTH and CBH), using the SGLI multiple channels of VNIR, SWIR, 763 nm, and TIR (Figs. 3 and 4).

The CBH retrievals derived from SGLI measurements using our algorithm exhibited quantitative consistency

with those measured using ceilometers. First, this study compared the SGLI CBH with the CBH collected via ground-based ceilometers deployed by the EUMETNET E-PROFILE (Fig. 5). The results demonstrated a high correlation ($r = 0.79$), small bias (approximately −80 m), and moderate RMSE (approximately 760 m) (Fig. 6), except in the case of multi-layer clouds (Fig. 7d–f). When the LCL was assumed to be an a priori estimate of CBH, the effectiveness of retrieving CBH from SGLI measurements was greater for mid- and high-level clouds

with a CBH of > 4 km than for low-level clouds. Moreover, this study also compared the SGLI CBH with the CBH measured using a ceilometer installed on the Antarctic research vessel *Shirase* (Fig. 8). Although the available data were limited to low CBH values, the results showed a strong correlation ($r = 0.82$) (Fig. 9).

The study also compared the zonal means of CBH and CTH obtained from the SGLI with those derived from a combination of CloudSat/CPR and CALIPSO/CALIOP observations (Fig. 10). Although the two zonal means of CBH exhibited good quantitative agreement, the zonal mean of the SGLI-derived CTH exhibited a systematic underestimation of at least 2 km. This can be attributed to the presence of cirrus clouds with a small COT and the associated multi-layer cloud structure. This underestimation of the CTH also suggests a systematic underestimation of the CGT by the SGLI. However, it should be emphasized that the CTHs obtained from the SGLI are in close agreement with those derived from the MOD06 product, which retrieves CTH from the MODIS TIR channels. This result highlights the well-known and persistent issue that CTHs derived from TIR are systematically lower than those detected by CALIOP, underscoring its critical impact on CGT estimation when combining the 763 nm and TIR channels.

The study concludes that CBH can be retrieved from actual measurements of the SGLI 763 nm channel, based on the results of these validations. This conclusion can facilitate the utilization of the SGLI 763 nm channel for cloud remote sensing and further encourage future satellite-borne passive sensors to be equipped with a 763 nm channel, in addition to the already promised VNIR, SWIR, and TIR channels for cloud property retrieval. Nevertheless, two issues remain regarding the retrieval of CBH and CGT data using the SGLI 763 nm channel. The first issue is whether our algorithm can extract all potential CBH and CGT information latent in the measurements of the SGLI 763 nm channel. The second issue is whether the combination of the 763 nm and TIR channels feasible with SGLI is optimal for retrieving CBH and CGT compared to other combinations, such as two channels in the oxygen A-band of POLDER-3/PARASOL (Ferlay et al., 2010) and two channels in the oxygen A-band and B-band of EPIC/DSCOVR (Davis et al., 2018). Addressing these issues will lead to a more comprehensive understanding of remote sensing that utilizes measurements in the oxygen absorption band to retrieve cloud geometrical parameters.

### 5.2 Implications and limitations for applications

The use of CBH retrievals derived from SGLI 763 nm measurements for estimating downward LW radiative flux is a promising application. The small difference between the CBHs derived from the SGLI and CPR-CALIOP observations, as depicted in Fig. 10, encourages the estimation of the downward LW flux from the SGLI CBH. In addition to CBH, the SGLI-derived COT, CER, ICOTF, and CTH, as illustrated in Figs. 3 and 4, can be utilized as inputs to estimate the upward/downward SW/LW radiative fluxes. This indicates that cloud radiative effects can be estimated from SGLI alone as satellite-based information, together with ancillary meteorological data. This application aligns with the objectives of the GCOM-C mission, which aims to better quantify the radiative effects of clouds by obtaining multiple cloud parameters.

The underestimation of the SGLI CTH in comparison with the CALIOP observations (Fig. 10) is likely to introduce bias into the upward LW flux estimation using the SGLI CTH as input. The difficulty lies in the fact

that the retrieved CTH can explain the observed radiance at the SGLI TIRs. Furthermore, SGLI has only two TIR channels used in our retrieval (10.8 and 12 μm), and additional TIR channels cannot be added to improve the current CTH retrieval. Therefore, the simplest ad hoc approach would be to empirically correct for the negative bias of the SGLI CTH when used as an input to the upward LW estimation.

The considerable uncertainty in the SGLI CGT retrievals for low-level clouds may limit their application. For example, the SGLI CGT retrievals can be combined with COT and CER to assess the extent of non-adiabatic processes in stratocumulus clouds, as demonstrated by Merk et al. (2016), who employed CGTs derived from radar observations. However, the moderate RMSE (480 m) of the SGLI CBH, shown in Figs. 6 and 9, are naturally propagated to CGT. This uncertainty should be considered when analyzing low-level clouds with cloud top heights of at most 2 km.

The handling of multi-layer cloud pixels is a remaining issue that requires immediate attention, as it introduces bias into SGLI CTH/CBH retrievals. Fortunately, previous studies on multi-layer cloud flags are plentiful and provide a variety of implementations, including pixel- and texture-based methods, which can be explored for their potential application to SGLI measurements.

The supplementary material also presents additional analysis results that may help in examining the issues that remain in our retrieval algorithm. Figure S5 demonstrates that the deviations in the zonal mean for all cloud properties retrieved by our algorithm were relatively minor with respect to the satellite zenith angle, indicating that the angle dependence of the algorithm is not significantly influential. However, the moderate angle-dependent variations in the zonal mean of CBH at low latitudes may require further investigation. Moreover, the series of analyses shown in Figures S6 - S8 illustrate that our algorithm can be influenced by a priori information regarding the state vector and the measurement vector, particularly the settings of $\boldsymbol{x}_a, \boldsymbol{S}_a$, and $\boldsymbol{S}_e$. It remains a matter for future research to investigate how to objectively determine the optimal settings for these a priori parameters.

*Data availability.* The GCOM-C/SGLI products for top-of-atmosphere radiation, cloud flags, cloud properties, and land surface reflectance, referred to as LTOA, CLFG, CLPR and RSRF products, respectively, are available online on the Globe Portal System (G-Portal) of JAXA (https://gportal.jaxa.jp). The GSMaP products are also available online at the JAXA G-Portal. The Modern-Era Retrospective Analysis for Research and Applications

Version 2 (MERRA-2) product are available online through the Goddard Earth Sciences Data and Information Services Center (http://disc.sci.gsfc.nasa.gov/mdisc/). The Group for High Resolution Sea Surface Temperature Level 4 (GHRSST) dataset (Chin et al., 2017) was supplied by the Physical Oceanography Distributed Active Archive Center at https://podaac.jpl.nasa.gov/dataset/MUR25-JPL-L4-GLOB-v04.2. The Terra Advanced Spaceborne Thermal Emission and Reflection Radiometer (ASTER) Global Digital Elevation Model (GDEM)

Version 2 (ASTGTM v002) is available online at NASA's Earth Observing System Data and Information System (EOSDIS) at https://doi.org/10.5067/ASTER/ASTGTM.002. The ground-based ceilometer data produced by the EUMETNET E-PROFILE are available online at the CERA Archive (https://archive.ceda.ac.uk/). The 2B-CLDCLASS-LIDAR and MOD06-1KM-AUX products are available online at The CloudSat Data Processing Center website (https://www.cloudsat.cira.colostate.edu).


*Author contribution.* TMN: Formal Analysis, Methodology, Software, Visualization, Writing – original draft; KS: Conceptualization, Funding acquisition, Writing – review & editing; MK: Methodology, Investigation, Writing – review & editing. All authors have read and agreed to the published version of the manuscript.

*Competing interests.* The authors declare that they have no conflicts of interest.

*Acknowledgments.* The authors are grateful to Luca Lelli and four anonymous reviewers for their invaluable comments that greatly helped improve this paper. This study was supported by the JAXA/GCOM-C project, the JAXA/EarthCARE project, and the Arctic Challenge for Sustainability II (ArCS II) Program (Grant

JPMXD1420318865). The authors are grateful to those who dedicated to the 59th-61st Japanese Antarctic Research Expedition (JARE 59-61). We thank the OpenCLASTR project for the use of the RSTAR7 package. We would also like to thank Takashi Y. Nakajima, Husi Letu, Miho Sekiguchi, and Hiroshi Ishimoto for providing the non-spherical ice particle scattering modules for the RSTAR7 package. Finally, we would like to thank Editage (www.editage.jp) for English language editing.

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
