# Peer review of "Retrieving cloud base height and geometric thickness using the oxygen A-band channel of GCOM-C/SGLI"

_Atmospheric Measurement Techniques, 2024_

## Author Comment (AC1)

First and foremost, we would like to express our sincere gratitude to Luca Lelli, the anonymous reviewers, the editor, and the editorial support team for taking the time to review our manuscript and provide valuable feedback. The comments we received were extremely helpful in improving our manuscript, and we are very grateful for them. As outlined below, we have revised the manuscript based on the feedback. The reviewers' comments are copied below and shown in *italics*, while our responses and the corresponding text in the manuscript are shown in red and orange, respectively.

**Response to the editorial support team**

*Regarding figures 3, 7: please ensure that the colour schemes used in your maps and charts allow readers with colour vision deficiencies to correctly interpret your findings. Please check your figures using the Coblis – Color Blindness Simulator (https://www.color-blindness.com/coblis-color-blindness-simulator/) and revise the colour schemes accordingly with the next file upload request.*

Answer: In response to the comment, we updated the color scheme for Figures 3 and 4 (excluding Figure 3a) to the 'Scientific Color Maps' recommended on the AMT submission page (https://www.atmospheric-measurement-techniques.net/submission.html).We recognize that adjusting the color scheme of the RGB images in Figures 3a and 7 as well would also be preferable. However, since the values of the three channels are directly assigned to R, G, and B, we are unsure how to modify them to make them colorblind-friendly. Instead, we utilized the 'Coblis – Color Blindness Simulator' to confirm that the RGB images in Figures 3 and 7 can be correctly interpreted by readers with anomalous trichromacy.

**Response to Luca Lelli**

*I read with interest this good article demonstrating the possibility of inferring cloud base height from a single channel in the oxygen absorption band as measured by SGLI but also with the support of multispectral measurements across the e/m spectrum.*

*It is not my intention with this commentary to provide a full review of the article or to judge the maturity of the work for possible publication. Since I myself am active in remote sensing of cloud properties, I would like to bring the following points to the authors' attention.*

Answer: We would like to thank you very much for carefully reading our manuscript and providing us with valuable comments. We have revised our manuscript, by taking full account of your suggestions. The original comments are copied below and shown in *italics*, while our responses and the corresponding text in the manuscript are shown in red and orange, respectively.

*In the introductory paragraph, at lines 53-65, there are two inaccuracies. This paragraph cites past work that "derive CBH and CGT using satellite-based passive instruments instead of active instruments" (line 53-54).*

*The Desmons et al (2019) citation at line 59 is incorrect. In that paper, an algorithm is presented that analyzes the sensitivity of the oxygen B-band centered around 688 nm to changes in cloud fraction and cloud pressure. By "cloud pressure", however, is meant a generic pressure (or height, once this value is converted with the help of an atmospheric profile) located at about the midpoint of the cloud body. The physical reasons are well known, namely that in the forward model of the algorithm the clouds are modeled not as real scattering bodies, but as Lambertian diffusers, for which light is not allowed to penetrate the clouds. But if the process of the photon penetration within a cloud is neglected, then any increase of the oxygen absoprtion line is interpreted as an existence of a cloud at a level that is lower that the actual altitude. This is a feature of the algorithm presented in Desmons et al (2019) and appropriate referenceses therein. In summary, the consequence of this assumption is that it is not possible for the algorithm to approximate multiple scattering inside the clouds, consequently it is not possible to derive any information about the height of the base of the clouds themselves. The authors in Desmons et al (2019), moreover, make no mention of any attempt to find information about CBH or CGT.*

*The Desmons et al, 2019, reference cannot be cited in the context of the retrieval of CBH nor CGT.*

Answer: We appreciate your pointing out the inaccurate citations in our manuscript and providing such a detailed explanation. We have double checked the literature in the light of your explanation and have come to understand our own error. Consequently, in accordance with your comment, we have removed the phrase "or B-band from the Global Ozone Monitoring Experiment (Desmons et al., 2019)" from the relevant paragraph. However, we believe that this paper is a significant contribution to the field of remote sensing using oxygen absorption channels. Therefore, we have added a new paragraph immediately after citing this and similar papers, as follows:

[Section 1; Lines 69 - 75]

"In addition to the literatures cited in the previous paragraph, there are several earlier studies that have investigated or attempted remote sensing of cloud geometric information using oxygen absorption channels while not retrieving both CBH and other geometric information (e.g. thickness). Examples include the use of oxygen A-band measurements (O'Brien and Mitchell, 1992), oxygen B-band measurements from the Global Ozone Monitoring Experiment (Desmons et al., 2019), and two channels in the oxygen A-band and B-band of the Earth Polychromatic Imaging Camera (EPIC) on the Deep Space Climate ObserVatoRy (DSCOVR) (Davis et al., 2018a; 2018b)."

*The second clarification I would like to bring to the authors' attention concerns the quote from Rozanov and Kokhanovsky, 2004 at line 65.*

*In that article, a set of Global Imager (GLI) and MERIS measurements is indeed analyzed, but the algorithm is concerned with the feasibility of deriving CTH and CBH (hence CGT) at the spectral resolution characteristic of the GOME, GOME-2 and SCIAMACHY family of instruments. Application of the algorithm, based this time on a realistic model of clouds composed of Mie droplets and a Gamma distribution, can be found in Rozanov and Kokhanovsky (2006) for GOME on ERS-2 and in Lelli and Vountas (2018) for SCIAMACHY on Envisat. In the second paper (Figure 3 and Table 1), the authors will find climatological values of CBH derived from SCIAMACHY directly comparable to their Figure 10 (page 23).*

*V. V. Rozanov and A. A. Kokhanovsky, "Determination of cloud geometrical thickness*

*using backscattered solar light in a gaseous absorption band," in IEEE Geoscience and Remote Sensing Letters, vol. 3, no. 2, pp. 250-253, April 2006, doi: 10.1109/LGRS.2005.863388*

*Lelli, L. and Vountas, M., 2018. Aerosol and cloud bottom altitude covariations from multisensor spaceborne measurements. In Remote Sensing of Aerosols, Clouds, and Precipitation (pp. 109-127). Elsevier. http://dx.doi.org/10.1016/B978-0-12-810437-8.00005-0*

Answer: Following the comment, we have declined to cite Rozanov and Kokhanovsky, 2004, and instead cite the two references you provided as follows:
[Section 1; Lines 64 - 69]
", and coincident measurements from two sensors providing the oxygen A-band and TIR channels, respectively: the Global Ozone Measurement Experiment (GOME) spectrometer and the Along Track Scanning Radiometer-2 (ATSR-2) on the European Remote-Sensing Satellite-2 (ERS-2) (Rozanov and Kokhanovsky, 2006); the SCanning Imaging Absorption spectroMeter for Atmospheric CHartographY (SCIAMACHY) and the Advanced Along-Track Scanning Radiometer (AATSR) onboard the European Environmental Satellite (Envisat) (Lelli and Vountas, 2018)."

Furthermore, the seasonal variation of the zonal mean CBH derived from GCOM-C/SGLI is illustrated in Figure S4 in the revised supplemental material. This figure is presented for comparison with Figure 3 of Lelli and Vountas (2018), which was derived from SCIAMACHY. The seasonal variation of the zonal mean of SGLI CBH (top) and the difference between JJA and DJF (bottom) are qualitatively consistent with those of SCIAMACHY CBH reported by Lelli and Vountas (2018). However, the CBH values from SGLI are roughly 1 km higher than those from SCIAMACHY in the low-latitude zone, including the peak value. Investigating the factors contributing to these quantitative differences between SGLI-derived and SCIAMACHY-derived CBHs (for example, differences in algorithms, channels used, or dependence on cloud type), remains an important task for the future research.
Therefore, the following paragraph has been added to the main text as well:
[Last paragraph of Section 4.4; Lines 606 - 613]
"It is also noteworthy that the zonal mean CBH derived from SGLI was in agreement with that retrieved from hyperspectral measurements in the oxygen A-band by

SCIAMACHY. Figure S4 illustrates the seasonal variations of the SGLI-derived zonal mean CBH and the difference between the JJA and DJF months, which generally aligns with those previously reported by Lelli and Vountas (2018) for SCIAMACHY. However, the SGLI-derived zonal mean CBH was approximately 1 km higher than that from SCIAMACHY in the low-latitude zone, including the peak value. Further investigation into the factors contributing to these quantitative differences between SGLI-derived and SCIAMACHY-derived CBHs, such as differences in algorithms, channels used, or cloud type dependencies, remain an important task for the future research."

*At line 136 the authors cite Rozanov & Kokhanovsky (2004) again in the context of "using an oxygen A-band channel paired with a TIR channel" (line 135). The Rozanov & Kokhanovksy paper makes no mention of TIR channles for the retrieval of cloud properties, because it focuses on the reflectance at Vis/NIR wavelenghts.*

Answer: Following the comment, we have removed the reference to "Rozanov & Kokhanovsky (2004)".

*This comment naturally leads me to ask the following question, also in light of the concepts presented by the authors in section 4.1 (Potential uncertainty in CBH retrieval).*

*Clearly, the accuracy of CBH depends on the accuracy of TIR-derived CTH and COT. This is even more important because in reflection, the signal arriving at the satellite will be generated through a different radiation-matter interaction process than in the Vis-NIR, so there will be a difference in the depth of light penetration (i.e. water has a single scattering albedo tending to 1 in the oxygen spectral bands while it fluctuates between 0.6 and 0.4 in the thermal infrared).*

*It would be extremely interesting if the authors could provide a more quantitative assessment of the errors in coincident COT, CTH(TIR) and CBH(NIR) as preliminary provided in Figure 15 (page 5689) of our paper in ACP (Lelli et al. 2014). There, one can see that errors in CBH are roughly proportional to CTH(NIR) by a factor in range 1.5 - 2.5. This is systematic and well behaved when COT/CTH and CBH are both retrieved in Vis/NIR. I am currently working on this issue and It is not known to me any error assessment in the case of a simultaneous and concurrent retrieval of COT/CTH from the*

*TIR and the CBH from the NIR.*

*Lelli, L., Kokhanovsky, A. A., Rozanov, V. V., Vountas, M., and Burrows, J. P.: Linear trends in cloud top height from passive observations in the oxygen A-band, Atmos. Chem. Phys., 14, 5679–5692, https://doi.org/10.5194/acp-14-5679-2014, 2014*

Answer: In the supplemental material, we have included a section (Text S2) on sensitivity analysis based on error propagation theory and radiative transfer simulation, together with the results shown in Figures S1 and S2, which we believe provide a response to this comment. Figure S1 demonstrates how perturbations in SW1, SW4, SW3, TI1, and VN9 propagate to the retrievals of COT, CER, ICOTF, CTH, and CBH. Figure S2 illustrates how uncertainties in the measurement vector induce uncertainties in the retrievals. Figures S2a and S2b compare different combinations of channels used in the retrieval process.

First, Figures S2a4 and S2a5 (or S2b4 and S2b5) show that the uncertainty of CBH was proportional to that of CTH by a factor ranging from 4 to 5, which is significantly larger than what Lelli et al. (2014) reported. Another notable feature is that, as shown in the first vertical panels (1-5, 1) of Figure S1, perturbations in SW1 induce not only COT error but also CBH error. This is likely to be a source of the larger uncertainty in CBH, observed in Figure S2. Additionally, the comparison of Figures S2a and S2b offers another important insight: incorporating an additional channel which provides COT information into the retrieval (here, VN11) can reduce not only the uncertainty in COT retrieval (Figure S2a1 → b1) but also the uncertainty in CBH retrieval (Figure S2a5 → b5). Note that the algorithm presented in the main text was performed with the better channel combination shown in Figure S2b, as described in Section 2.2.1.

Therefore, the following paragraph has been added to the main text as well:

[Last paragraph of Section 4.1; Lines 412 - 425]

"In our algorithm, the uncertainty in CBH retrieval is also entangled with the uncertainty in COT retrieval. We performed a sensitivity analysis based on the error propagation theory to examine how measurement uncertainties propagate to retrieval uncertainties (see Text S2 in the supplemental material). Figure S1 demonstrates how perturbations in individual measurement channels induce retrieval errors. Notably, perturbations in SW1, which is a channel sensitive to COT but located outside the oxygen A-band, can induce errors not only in COT retrieval (Fig. S1(1,1)) but also in CBH retrieval (Fig. S1(5,1)). This indicates that COT errors disturb the separation of COT and CBH from VN9 measurements. Figure S2 further demonstrates how the overall uncertainty in the multiwavelength measurements incorporated into the inverse estimation propagates to retrieval uncertainties. The comparison of Figs. S2a1 and S2b1 reveals incorporating VN11 alongside SW1 reduce the uncertainty in COT retrieval, which, in turn, contributes to reduce uncertainty in CBH retrieval. As described in Section 2.2.1, our algorithm utilized both SW1 and VN11. The results of these sensitivity analyses emphasize the importance of carefully addressing uncertainties in COT retrieval when deriving CBH from VN9 measurements. The entanglement of COT, CTH, and CBH retrieval errors associated with oxygen A-band measurements has also been reported by Lelli et al. (2014)."

---

## Author Comment (AC3)

First and foremost, we would like to express our sincere gratitude to Luca Lelli, the anonymous reviewers, the editor, and the editorial support team for taking the time to review our manuscript and provide valuable feedback. The comments we received were extremely helpful in improving our manuscript, and we are very grateful for them. As outlined below, we have revised the manuscript based on the feedback. The reviewers' comments are copied below and shown in *italics*, while our responses and the corresponding text in the manuscript are shown in red and orange, respectively.

**Response to the editorial support team**

*Regarding figures 3, 7: please ensure that the colour schemes used in your maps and charts allow readers with colour vision deficiencies to correctly interpret your findings. Please check your figures using the Coblis – Color Blindness Simulator (https://www.color-blindness.com/coblis-color-blindness-simulator/) and revise the colour schemes accordingly with the next file upload request.*

Answer: In response to the comment, we updated the color scheme for Figures 3 and 4 (excluding Figure 3a) to the 'Scientific Color Maps' recommended on the AMT submission page (https://www.atmospheric-measurement-techniques.net/submission.html).We recognize that adjusting the color scheme of the RGB images in Figures 3a and 7 as well would also be preferable. However, since the values of the three channels are directly assigned to R, G, and B, we are unsure how to modify them to make them colorblind-friendly. Instead, we utilized the 'Coblis – Color Blindness Simulator' to confirm that the RGB images in Figures 3 and 7 can be correctly interpreted by readers with anomalous trichromacy.

**Response to Anonymous Referee #3**

*A new algorithm was proposed to retrieve the CBH using the SGLI 763 nm channel in combination with several other SGLI channels in the visible, shortwave infrared, and thermal infrared regions. However, there are some critical aspects that require more detailed elaboration and clarification to enhance the clarity of your findings.*

We would like to thank you very much for carefully reading our manuscript and providing us with valuable comments. We have revised our manuscript, by taking full account of the referee's suggestions. The original comments are copied below and shown in *italics*, while our responses and the corresponding text in the manuscript are shown in red and orange, respectively.

1. *The optimal estimation algorithm is undoubtedly the heart of your study. Please expand on the methodology section to provide a comprehensive and step-by-step description of the algorithm. This should include the mathematical formulations, assumptions made, and any pre-processing or post-processing steps involved. This will enable readers to fully understand your work.*

Answer: In accordance with the comment, the description of the optimal estimation method using the Levenberg–Marquardt (LM) approach has been revised and elaborated as follows in Section 2.2.1 of the revised manuscript:

[Section 2.2.1; Lines 194 - 209]

"Our algorithm employs the optimal estimation method framework (Rodgers, 2000), which seeks the optimal solution that minimizes the cost function $J$ that is given by the following equation:

$$J = [y - F(x)]^T S_e^{-1} [y - F(x)] + [x - x_a]^T S_a^{-1} [x - x_a], \qquad (1)$$

where $y$ and $F(x)$ represent the measurement vectors consisting of the measured and simulated TOA reflectances, respectively. $x$ represents the state vector, $x_a$ represents the a priori values for $x$, and $S_e$ and $S_a$ represent the covariance matrices for $y$ and $x_a$, respectively.

The iterative solution of the inverse problem through the Levenberg–Marquardt approach, based on the Gauss-Newton method for minimizing Equation (1), is determined using the following formula (Rodgers, 2000):

$$x_{i+1} = x_i + [(1+\gamma)\,S_a^{-1} + K_i^T S_e^{-1} K_i]^{-1}\{K_i^T S_e^{-1}[y - F(x_i)] - S_a^{-1}[x_i - x_a]\}, \quad (2)$$

where $x_i$ is the state vector at the $i$-th iteration, $K_i$ is the Jacobian matrix of $F(x)$ evaluated at $x_i$, $\gamma$ is a parameter chosen at each step of the iteration to reduce the cost function $J$."

2. *Clarify how you address the non-Gaussian distributions of the observations. Discuss the limitations and potential biases introduced by these assumptions.*

Answer: The optimal estimation method (Rodgers, 2000) in Formula (1) is formulated based on the assumption that the a priori distributions for the measurement vector and state vector (i.e., $S_e$ and $S_a$) follow Gaussian distributions. Consequently, non-Gaussian distributions are not considered in this study.

We believe it is reasonable to assume a Gaussian distribution for measurement errors of TOA radiances (i.e., $S_e$) that has been accurately calibrated radiometrically and exhibits sufficiently small measurement errors. However, for the prior distribution of the state vector (i.e., $S_a$), non-Gaussian distributions may be more appropriate in certain circumstances. For instance, the log-normal distribution might better fit the histogram of cloud optical thickness. However, since this is the first application of our algorithm to GCOM-C/SGLI, we used a normal distribution for simplicity with means of typical orders of magnitude and fairly large standard deviations to avoid excessive reliance on the prior distribution. To clarify these points, the following text has been added.

[Section 2.2.1; Lines 221 - 225]

"Note that the values in Table 1 could be assigned more appropriate prior distributions (mean, standard deviation, and even covariance) by using cloud property products from other satellite observations. However, since this is the first application of our algorithm to GCOM-C/SGLI, we used a normal distribution for simplicity with means of typical orders of magnitude and fairly large standard deviations to avoid excessive reliance on the prior distribution."

3. *Provide details on how you estimate and incorporate the covariance matrix of the observations, particularly addressing the correlations between different channels. Discuss any challenges in estimating these correlations and the strategies employed*

*to mitigate their impacts on the estimation accuracy.*

Answer: For the covariance matrix of the measurement vector $S_e$, the diagonal elements (i.e., variances), which represent uncertainties of measurements for each channel, were given based on the post-launch calibration information of the GCOM-C mission. Meanwhile, the non-diagonal elements (i.e., covariances), which represent correlation of measurement errors between channels, were all set to zero. To clarify these points, we have revised the text as follows:
[Section 2.2.1; Lines 216 - 218]
"The diagonal elements of $S_e$, consisting of the uncertainties in the TOA measurements, were obtained from the post-launch calibration information of the GCOM-C mission, while the non-diagonal elements of $S_e$ were set to zero."

In general, issues that cause significant bias across multiple channels (e.g. electrical leakage and/or stray light) are identified and addressed during the sensor development stage. For GCOM-C/SGLI, no such critical problems have been reported since its launch. Additionally, radiance calibration after launch is typically performed independently for each channel, providing the uncertainty (i.e., variances) of the radiance for each channel, but it does not usually provide the correlation (i.e., covariance) of the measurement errors between channels. It may be possible to estimate the covariance of measurement errors through vicarious calibration using in-situ measurements or mutual comparisons with other satellite sensors; however, the reliability of such estimates is low. In conclusion, we consider it challenging to assign a value to the correlation of measurement errors for $S_e$ and deem it most appropriate to assume it as 0.

4. *Explain how you account for angular biases or other systematic errors in the observations, particularly as they relate to the state variables.*

Answer: In the optimal estimation method, systematic biases such as angular biases are typically addressed by adding them collectively to $S_e$, rather than considering them individually. However, if all possible systematic biases were to be directly added together, $S_e$ will become too large relative to $S_{a'}$, causing the solution to be overly constrained to $x_a$. Therefore, as described in the main text, we determined $S_e$ based on the post-launch calibration information for GCOM-C, while assigning relatively loose prior distributions to $S_a$ as shown in Table 1, and designed the algorithm to be tightly

constrained by the SGLI measurements rather than by the a priori information.

5. *Elaborate on the methodology used to determine the background error covariance for the state variables. Specifically, discuss how you handle correlations between different state variables and how you arrived at the values presented in Table 1. Consider discussing the sensitivity of your results to these assumptions and any validation performed to support the chosen values.*

Answer: Please allow us to partially reiterate our response to Comment 2 above in addressing this comment. The values in Table 1 were provided roughly based on cloud property products from other satellite observations, without overly constraining the solution space. For the optimal estimation method to be most effective, a prior distribution close to the true value should be used. However, since this is the first application of our algorithm to GCOM-C/SGLI, we used a normal distribution with means of typical orders of magnitude and fairly large standard deviations to avoid excessive reliance on the prior distribution. To clarify these points, the following text has been added in the revised manuscript.

[Section 2.2.1; Lines 221 - 225]

"Note that the values in Table 1 could be assigned more appropriate prior distributions (mean, standard deviation, and even covariance) by using cloud property products from other satellite observations. However, since this is the first application of our algorithm to GCOM-C/SGLI, we used a normal distribution for simplicity with means of typical orders of magnitude and fairly large standard deviations to avoid excessive reliance on the prior distribution."

6. *Detail how you estimate the uncertainty in cloud-base height (CBH) from your optimal estimation algorithm. This should include a discussion of the error propagation and any assumptions made in the uncertainty analysis.*

Answer: As the reviewer has pointed out, based on the error propagation theory (employing the optimal estimation framework, the covariance matrix $S_e$, and the forward calculation), it is possible to estimate the uncertainty in the CBH estimation. However, the uncertainty in CBH estimated in this manner is typically underestimated because the primary sources of estimation error arise from factors not accounted for in the forward

calculation, such as the vertical inhomogeneity of the cloud property profile and multilayer cloud structures. Therefore, this study assessed the uncertainty in CBH through validation against CBH measurements obtained from ground-based and ship-borne ceilometers (Figures 6 and 9), as well as satellite-borne radar and lidar (Figure 10). In fact, the bias in CTH (Figure 9) and the dependence of CBH bias on multi-layer cloud structure (Figures 6 and 7) observed from these validation analysis are not captured by estimates based solely on the error propagation theory.

7. *Consider performing a sub-analysis by classifying clouds into different types and reporting the results separately. This would help isolate the impacts of cloud type on your findings and provide valuable insights into the variability in estimation performance across cloud types.*

Answer Thank you for your valuable suggestion. We agree that it is important to investigate in more detail how our algorithm behaves for different cloud types. However, we do not have an 'objective information/method' that decomposes the validation results shown in Figures 6, 9, and 10 into different cloud types. For instance, the all-sky camera used to identify cloud types is not necessarily operated at the same time as the ceilometers. Additionally, although observations from CloudSat and CALIPSO are effective for identifying cloud types, they are unfortunately not collocated with SGLI, except in high-latitude regions, because the A-Train satellites operate in the afternoon orbit while GCOM-SGLI operates in the morning orbit. Although it is possible to apply COT and CTH retrieved from SGLI to the traditional cloud type classification of the ISCCP, this classification itself is dependent on the retrieval errors of COT and CTH themselves, and the uncertainty in cloud type classification may not be fully resolved. One potential approach could be to use cloud type classification from geostationary meteorological satellites as an objective third-party. However, such analysis is beyond the scope of this study and we would like to leave it as a topic for future work.

8. *Some new cloud base height retrieving method should be cited in the Introduction. such as: Retrieving cloud base height from passive radiometer observations via a systematic effective cloud water content table, Remote Sensing of Environment, 294 (2023), 113633.*

Answer: Thank you for your valuable suggestion of a literature. We have incorporated it into both the Introduction and References as follows:

[Section 1; Lines 76 - 79]

"Another approach is to estimate other cloud properties correlated with CBH and CGT, such as CTH and COT, which are usually measured using passive instruments not equipped with oxygen absorption channels. This approach has been implemented using adiabatic (Seaman et al., 2017) or statistical models (Noh et al., 2017, 2022; Shao et al., 2023; Tan et al., 2023)."

[References]

"Tan, Z., Ma, S., Liu, C., Teng, S., Letu, H., Zhang, P., and Ai, W.: Retrieving cloud base height from passive radiometer observations via a systematic effective cloud water content table, Remote Sens. Environ., 294, 113633, https://doi.org/10.1016/j.rse.2023.113633, 2023."

---

## Author Comment (AC4)

First and foremost, we would like to express our sincere gratitude to Luca Lelli, the anonymous reviewers, the editor, and the editorial support team for taking the time to review our manuscript and provide valuable feedback. The comments we received were extremely helpful in improving our manuscript, and we are very grateful for them. As outlined below, we have revised the manuscript based on the feedback. The reviewers' comments are copied below and shown in *italics*, while our responses and the corresponding text in the manuscript are shown in red and orange, respectively.

**Response to the editorial support team**

*Regarding figures 3, 7: please ensure that the colour schemes used in your maps and charts allow readers with colour vision deficiencies to correctly interpret your findings. Please check your figures using the Coblis – Color Blindness Simulator (https://www.color-blindness.com/coblis-color-blindness-simulator/) and revise the colour schemes accordingly with the next file upload request.*

Answer: In response to the comment, we updated the color scheme for Figures 3 and 4 (excluding Figure 3a) to the 'Scientific Color Maps' recommended on the AMT submission page (https://www.atmospheric-measurement-techniques.net/submission.html).We recognize that adjusting the color scheme of the RGB images in Figures 3a and 7 as well would also be preferable. However, since the values of the three channels are directly assigned to R, G, and B, we are unsure how to modify them to make them colorblind-friendly. Instead, we utilized the 'Coblis – Color Blindness Simulator' to confirm that the RGB images in Figures 3 and 7 can be correctly interpreted by readers with anomalous trichromacy.

**Response to Anonymous Referee #4**

*The submission by Nagao et al. combines a 4-channel 4-property cloud retrieval with a cloud phase differentiation to get liquid and ice cloud properties output in a single product. This is a neat idea and I found the paper pretty well structured and easy to read. I appreciate the authors found whatever validation data they could and I liked the comparison of cloud base heights (CBH) with reanalysis lifting condensation level.*

We would like to thank you very much for carefully reading our manuscript and providing us with valuable comments. We have revised our manuscript, by taking full account of the referee's suggestions. The original comments are copied below and shown in *italics*, while our responses and the corresponding text in the manuscript are shown in red and orange, respectively.

*I have a few comments I would request the authors address but I would be happy to support publication following improvements. My comment areas can be summarised as: (1) fixing some issues in the literature review and process description, (2) clarifying some method details, (3) expanding slightly on the verification step, (4) doing some quick theoretical error quantification calculations.*
*I don't believe my requests would greatly change the results or conclusions, but would improve the quality of the manuscript.*

*1. LITERATURE REVIEW AND BACKGROUND*
*Beginning L53 I read it as talking about studies that retrieved BOTH cloud top heights (CTH) and other geometric information (e.g. thickness), but it appears Desmons et al. (2017) gets a single pressure and Davis et al. (2018) talks specifically about how DISCOVR can only get one piece of vertical information. The latter is a two-part analysis and I think both are worth citing, including doi: 10.1016/j.jqsrt.2018.09.006 . Either rephrase, remove these citations, or see my comments a couple of paragraphs below here to see where they could fit.*

Answer: We have removed Desmons et al. (2017) and Davis et al. (2018) from the relevant paragraph. However, we believe that these papers, along with the other paper by Davis et al. that you mentioned newly, are significant contributions to the field of remote sensing using oxygen absorption channels. Therefore, we have added a new paragraph immediately after citing these three papers, as follows:

[Section 1; Lines 69 - 75]

"In addition to the literatures cited in the previous paragraph, there are several earlier studies that have investigated or attempted remote sensing of cloud geometric information using oxygen absorption channels while not retrieving both CBH and other geometric information (e.g. thickness). Examples include the use of oxygen A-band measurements (O'Brien and Mitchell, 1992), oxygen B-band measurements from the Global Ozone Monitoring Experiment (Desmons et al., 2019), and two channels in the oxygen A-band and B-band of the Earth Polychromatic Imaging Camera (EPIC) on the Deep Space Climate ObserVatoRy (DSCOVR) (Davis et al., 2018a; 2018b).
"

*On L124 you state "Therefore, for a given CTH, the amount of oxygen above the clouds is uniquely determined", but isn't it for cloud-top pressure?*
*Next up, I would suggest using the path-length framing for parts of your description as you judge appropriate. Lower or thicker clouds = photons travel further and there is more absorption by O2. Then explain how it is hard to work out how much absorption is above versus within cloud, which is why your retrieval relies heavily on another source for CTH. I personally find this framing much simpler and clearer and I think some readers who are not A-band experts will appreciate it.*

Answer: Thank you for your thoughtful suggestion. We have revised the first few paragraphs of Section 2.2.1, which explains how the oxygen A-band channel is sensitive to variations in both CTH and CGT, by incorporating your proposal as follows:
[Section 2.2.1; Lines 124 - 147]
"The top-of-atmosphere (TOA) reflectance in the oxygen A-band channel (~763 nm), which is characterized as moderate to strong oxygen absorption, exhibits sensitivity to both CTH and CGT. This mechanism can be described as follows. First, for simplicity, we assume a black surface with an albedo of zero, gas absorption only by oxygen, and a geometrically thin (CGT~0) but optically reasonably thick (COT > 0) plane-parallel cloud. Additionally, we assume that the cloud properties used to parameterize cloud scattering— namely, COT, CER, and cloud thermodynamic phase, are known. Under these conditions, the TOA reflectance in visible and near-infrared channels outside the oxygen absorption band (e.g., the VN9, VN11, and SW1 channels of SGLI, centered at 673 nm, 868 nm, and 1.05 μm, respectively) can be accurately represented using these cloud property parameters.

In the oxygen absorption channel, sunlight is significantly absorbed by the oxygen above

the clouds before and after being reflected by the clouds on its path to the satellite. The TOA reflectivity in the oxygen absorption channel can be expressed with two additional parameters: CTH and the amount of oxygen above CTH. Conveniently, oxygen is well-mixed in the atmosphere, and its mixing ratio can be assumed to be globally constant and known. Thus, if the CTH (or cloud top pressure, equivalently) is given, the amount of oxygen above cloud can be immediately calculated. Therefore, when CGT~0, it is sufficient for parameterizing the TOA reflectance in the oxygen absorption channel to have CTH in addition to COT, CER, and cloud thermodynamic phase.

When CGT > 0, the CGT (or CBH) needs to be included as an additional parameter for explaining the TOA reflectance in the oxygen absorption band. When the cloud is geometrically thickened without changing the CTH (i.e., when CGT increases and CBH decreases), the TOA reflectance should decrease owing to oxygen absorption within the cloud layer between CTH and CBH. This is because, as the CBH decreases, sunlight travels a longer distance in the cloud layer, increasing the opportunities for oxygen absorption. These explanations provide an intuitive understanding of why the TOA reflectance in the oxygen absorption channel is sensitive to variations in both CTH and CGT (or CBH)."

*Finally, on L131—135 you mention you would need a couple of channels to separate cloud-top and cloud-thickness, but isn't that assuming you already know other stuff like optical depth? Actually, I think you're underselling the difficulty of getting geometric thickness here and readers should know that you're attempting something that is very challenging. You can cite O'Brien and Mitchell (1992, doi: 10.1175/1520-0450(1992)031<1179:EEFROC>2.0.CO;2) and/or Richardson et al. (2018, doi: 10.5194/amt-11-1515-2018) as those papers show the challenging spectral resolution requirements for pure A-band approaches. The Desmons and Davis references could fit here.*

Answer: In response to the comment, we have revised the paragraph containing the sentence you mentioned as follows:
[Section 2.2.1; Lines 148 - 151]
"The retrievals of CTH and CGT from the oxygen absorption band through this principle can be facilitated by a pair of spectral channels with different sensitivities to these cloud parameters since the oxygen absorption channels are sensitive to both CTH and CGT. This is the case even when cloud properties other than CTH and CGT (i.e., COT, CER, and cloud thermodynamic phase) are known."

*1. METHODOLOGY DETAIL*

*I think you did a nice job here, being efficient in your explanations and I believe I could reproduce a lot of your retrieval thanks to details like Table 1. However, I haven't used RSTAR. Can you specify the gas absorption spectra source, including if/how line broadening is handled. Is there anything else that's relevant: e.g. effective spectral resolution or the angular resolution of your scattering calculations?*

Answer: We have added a text providing information about the gas absorption tables included in the RSTAR7 package:

[Section 2.2.2; Lines 234 - 237]

"The RSTAR version 7 (RSTAR7) package includes gas absorption line tables compiled into narrow bands using the k-distribution method. However, the k-distribution table is based on the HITRAN 2004 molecular spectroscopic database (Rothmana et al., 2005) and does not incorporate recent updates to the oxygen absorption lines."

*Could you also comment briefly on the cloud flag – how does it handle optically thin or broken clouds? I understand your retrieval assumes plane parallel, but a general statement on whether we should expect lots of broken clouds to be retrieved would help me to interpret things.*

Answer: Since the assumption is a plane parallel cloud structure, contamination of clear-sky regions at the sub-pixel scale near the cloud edge causes significant bias in cloud retrievals (Zhang and Platnick, 2011) In scenes with small cumulus clouds, which are prone to clear-sky contamination, quality flags based on indicators such as the spatial standard deviation of TOA reflectance in the VNIR channels or the retrieved COT would be useful. Although such flags have not been implemented in this study, it would be valuable to carry out such flagging if the target scene can be identified as primarily consisting of cumulus clouds.

- Zhang, Z. and Platnick, S.: An assessment of differences between cloud effective particle radius retrievals for marine water clouds from three MODIS spectral bands, Journal of Geophysical Research: Atmospheres (1984–2012), 116, https://doi.org/10.1029/2011jd016216, 2011.

*1. VALIDATION*

*When you compare against the ground-based ceilometers you use the local standard deviation and "empirically determined" some data selection. I don't think we can be sure that the result is appropriate to compare with your values estimated from the previous section, because you might have filtered for "best behaving" scenes. Is there a way to identify these "better" scenes from your satellite product? If not then you should explicitly state that this as a potential issue for the comparison.*

Answer: As the reviewer pointed out, only SGLI scenes with a local standard deviation of ceilometer CBH below 1 km were used for validation in Figure 6 of Section 4.2. We believe this data screening is essential for the following reasons. First, since ceilometer cannot perform spatially wide measurements, ensuring temporal stability would allow us to extract a spatially homogeneous cloud field. Additionally, this screening helps minimize the impact of local variability in ceilometer CBH on the RMSE of SGLI CBH relative to ceilometer CBH.

However, as you commented, this screening may inadvertently result in selecting scenes and cloud types that are more favorable for SGLI cloud retrievals. This is because factors that cause significant biases in cloud retrievals, such as sub-pixel scale heterogeneity of COT and CBH, are more likely to be common in scenes with diverse cloud types, such as cumulus and cirrus clouds, which are filtered out by this screening. Although it may be possible to identify these factors using indicators such as the spatial standard deviation of cloud optical thickness, this study does not implement such quality flags. To clarify this concern, we have revised the main text as follows:

[Section 4.2; Lines 443 - 451]

"This is because data with significant temporal variability are more likely to encompass diverse cloud types, such as cumulus and cirrus clouds. Notably, these three thresholds ($\Delta s < 4$ km, $\Delta t < 30$ min, and $SD[CBH_{ground-based}] < 1$ km) were empirically determined. However, it is important to note that the data screening using this $SD[CBH_{ground-based}] < 1$ km might unintentionally exclude scenes and cloud types that are challenging to retrieve. This is because factors that cause significant biases in cloud retrievals, such as sub-pixel scale heterogeneity of COT and CBH, are more likely to be common in scenes with diverse cloud types, such as cumulus and cirrus clouds, which are filtered out by this screening. Therefore, it should be noted that the SGLI cloud properties retrieved using our algorithm may contain lower-quality data than those

presented in the validation results here."

*Can you do the CloudSat comparison split by phase? It would be cool to see, but it's not worth much extra effort on your part so I am not pushing hard for this. If you have to reprocess all the granules to get your fields then don't bother - it's only a minor request.*

Answer: Thank you for the valuable suggestion. It is possible to classify CloudSat/CALIPSO-derived CTH/CBH using the cloud thermodynamic phase information identified from CALIPSO lidar measurements, included in the CloudSat 2B-CLDCLASS-LIDAR product. However, because the footprint of CALIPSO lidar and the swath of SGLI rarely overlap, except in high-latitude regions (CloudSat and CALIPSO operate in the afternoon orbit, while GCOM-C/SGLI operate in the morning orbit), the CALIPSO-derived cloud phase cannot be used for the classification of the SGLI CTH/CBH. Our algorithm also retrieves cloud phase information (ice COT fraction as shown in Figures 3d and 4c) utilizing the SWIR channels of SGLI, but it would capture the cloud phase for deeper optical depths compared to CALIOP, which is limited to detecting the cloud phase near the cloud tops (Nagao and Suzuki, 2022 JGR). As a third source capable of uniformly classifying CTH/CBH from both CloudSat/CALIPSO and SGLI, the use of cloud phase obtained from geostationary satellites, such as Himawari and GOES series, could be considered. However, such analysis is beyond the scope of this study and we would like to leave this as a subject for future work.

- Nagao, T. M. and Suzuki, K.: Characterizing Vertical Stratification of the Cloud Thermodynamic Phase With a Combined Use of CALIPSO Lidar and MODIS SWIR Measurements, J Geophys Res Atmospheres, 127, https://doi.org/10.1029/2022jd036826, 2022.

*1. UNCERTAINTY TESTS*
*We know that cloud properties correlate, but I'm fine with you sticking with a diagonal prior given the data we have available. Also, we know that the forward model also has error, which it seems you don't include in your optimisation. Therefore your retrieval might be too "tight" to the observations, right? I think these are legitimate concerns that would take a full other project to appropriately quantify. However, you could address them by re-running a subset of the retrieval footprints and evaluating how the retrievals*

*change. I request 3 tests*

Answer: Thank you for your valuable comment. We agree that in order to run the optimal estimation method most effectively, it is important to assign appropriate parameters, including correlations between cloud properties, to a prior distribution of the state vector ($\boldsymbol{S}_a$). We also understand that correlations between cloud properties, such as COT, CTH, and CBH, which are derived from satellite observations, can be utilized in $\boldsymbol{S}_a$. However, this paper represents one of the first applications of the GCOM-C SGLI oxygen A-band channel (VN9), and its primary objective is to demonstrate that CGT and CBH information can be extracted from SGLI VN9 measurements. Therefore, we assigned relatively loose prior distributions to $\boldsymbol{S}_a$, as shown in Table 1, and designed the algorithm to be tightly constrained by the SGLI measurements. While utilizing correlations between cloud properties derived from satellite observations in the optimal estimation method is a promising approach, we consider it as a challenge for future work.

- *Try with different priors to see whether your results are meaningfully sensitive to them. Shift means term-by-term, and/or scale the standard deviations. The most interesting would be to introduce a cross correlation e.g. between tau and CGT/CBH. Even if it's only a small correlation*

Answer: As mentioned above, assuming correlations between cloud properties (i.e., modifying the off-diagonal elements of $\boldsymbol{S}_a$) does not align with the purpose of this paper. Therefore, we have decided not to include it here. Instead, we tested the algorithm using a perturbed $\boldsymbol{x}_a$ and relaxed $\boldsymbol{S}_a$, and the results are presented in Figures S6 and S7, respectively, of the revised supplemental material. Figure S6 compares two cloud retrievals: the x-axis represents the original retrieval under the default settings described in the main text, while the y-axis represents the retrieval obtained when the elements of $\boldsymbol{x}_a$ were assigned random numbers following a uniform distribution within the ranges defined in Table 1. Figure S7 is similar to Figure S6, except the y-axis represents the retrieval obtained under a relaxed priori distribution of $2\boldsymbol{S}_a$. These figures demonstrate that perturbations to $\boldsymbol{x}_a$ and relaxations of $\boldsymbol{S}_a$ primarily increase the RMSEs of retrieved cloud properties, while not having a significant impact on the biases of them. Therefore, the following text has been added to the main text as well:

[Last paragraph of Section 5.2; Lines 691 - 695]

"Moreover, the series of analyses shown in Figures S6 - S8 illustrate that our algorithm

can be influenced by a priori information regarding the state vector and the measurement vector, particularly the settings of $x_a$, $S_a$, and $S_e$. It remains a matter for future research to investigate how to objectively determine the optimal settings for these a priori parameters."

- *A subset of simulations with $y$ elements perturbed in turn? The perturbation magnitudes should probably be proportional to the standard deviations implied by S_e for consistency. This would tell us something about how errors in each particular channel would feed through into the overall retrieval.*

Answer: In the supplemental material, we have included a section (Text S2) on sensitivity analysis based on error propagation theory and radiative transfer simulation with their results shown in Figures S1 and S2. We consider Figure S1 to be a direct response to this comment. Specifically, Figure S1 demonstrates how perturbations of signals in SW1, SW4, SW3, TI1, and VN9 propagate to the retrievals of COT, CER, ICOTF, CTH, and CBH. The horizontal panels in Figure S1 correspond to the measurement errors applied to specific channels, while the vertical panels correspond to the impact of these errors on retrievals of cloud properties. For example, Figure S1 (1-5,1) shows the COT error caused by the perturbation in SW1. One important takeaway from Figure S1 is that the perturbation in SW1 has a relatively large impact not only on COT but also on CBH. This suggests that the accuracy of COT is crucial for extracting CBH information from VN9. Therefore, the following text have been added to the main text as well:
[Last paragraph of Section 4.1; Lines 412 - 425]
"In our algorithm, the uncertainty in CBH retrieval is also entangled with the uncertainty in COT retrieval. We performed a sensitivity analysis based on the error propagation theory to examine how measurement uncertainties propagate to retrieval uncertainties (see Text S2 in the supplemental material). Figure S1 demonstrates how perturbations in individual measurement channels induce retrieval errors. Notably, perturbations in SW1, which is a channel sensitive to COT but located outside the oxygen A-band, can induce errors not only in COT retrieval (Fig. S1(1,1)) but also in CBH retrieval (Fig. S1(5,1)). This indicates that COT errors disturb the separation of COT and CBH from VN9 measurements. Figure S2 further demonstrates how the overall uncertainty in the multi-wavelength measurements incorporated into the inverse estimation propagates to retrieval uncertainties. The comparison of Figs. S2a1 and S2b1 reveals incorporating VN11 alongside SW1 reduce the uncertainty in COT retrieval, which, in turn, contributes to

reduce uncertainty in CBH retrieval. As described in Section 2.2.1, our algorithm utilized both SW1 and VN11. The results of these sensitivity analyses emphasize the importance of carefully addressing uncertainties in COT retrieval when deriving CBH from VN9 measurements. The entanglement of COT, CTH, and CBH retrieval errors associated with oxygen A-band measurements has also been reported by Lelli et al. (2014)."

- *Some tests with a relaxed S_e, just scale it to be larger. This would represent the effect of model error.*

Answer: We tested the algorithm using a relaxed $S_e$, with the results presented in Figures S8 of the supplemental material. Figure S8 compares two cloud retrievals: the x-axis represents the original retrieval under the default settings described in the main text, while the y-axis represents the retrieval obtained under an increased measurement uncertainty of $2S_e$. Figure S8 shows that COT and CER tend to be underestimated for clouds with large values of COT and CER. This is believed to occur because as COT and CER increase, the sensitivity decreases (with smaller changes in TOA reflectance), and as $S_e$ increases, the iteration toward the solution from the initial values tends to converge earlier. However, this underestimation of COT and CER does not have a significant impact on ICOTF, CTH, and CBH.

Therefore, the following text has been added to the main text as well:

[Last paragraph of Section 5.2; Lines 691 - 695]

"Moreover, the series of analyses shown in Figures S6 - S8 illustrate that our algorithm can be influenced by a priori information regarding the state vector and the measurement vector, particularly the settings of $x_a, S_a$, and $S_e$. It remains a matter for future research to investigate how to objectively determine the optimal settings for these a priori parameters."

*MINOR COMMENTS:*
*L280 – why the change in resolution? Why specified dates?*

Answer: Unfortunately, our algorithm is not fast enough to process the original SGLI LTOAK radiance product, which is projected onto a sinusoidal tile grid at a resolution of 1/120° (~1 km) and used in Figure 3, across the entire globe for a single year. To demonstrate an example of global processing shown in Figure 4, we used the SGLI

LTOAF product—an alternative radiance dataset provided by the GCOM-C mission, created by downsampling the LTOAK files to a resolution of 1/24°.

*L319—327: "it is not surprising" and associated text seems like a long winded way of saying that a cloud with 3 km cloud top cannot be 6 km thick… you could cut this down for brevity.*

Answer: In response to the comment, the text in the corresponding paragraph has been revised and shortened as follows:

[Section 3.2; Lines 363 - 366]

"The global distribution of CGT in Fig. 4f also agrees with typical climatological cloud regimes and was similar to that of CTH. For example, mean CGTs of less than 2 km were observed in cumulus cloud regions within the trade-wind zone and in stratocumulus decks over the ocean off the western coasts of continents, whereas mean CGTs greater than 6 km corresponded to deep convective clouds in the tropics."

*L407: "SGLI, VN3, SW3, and SW4 channels" I think the first comma should be removed to read "SGLI VN3, SW3, and SW4 channels". The current writing makes it seem like SGLI is part of the list.*

Answer: We have removed "," after SGLI.

*L481—484: this bit just seems obvious. You could lose the last two sentences.*

Answer: In response to the comment, the text in the corresponding paragraph has been revised and shortened as follows:

[Section 4.4; Lines 547 - 550]

"This section presents a comprehensive validation of both CTH and CBH derived from SGLI, complementing the validation of CBH alone presented in the previous two sections. As illustrated in Fig. 2, the accuracy of CBH retrieval using the 763 nm channel is shown to depend on the accuracy of CTH retrieval. Therefore, in addition to the CBH-only validation, a thorough validation of both CTH and CBH retrievals is necessary."

*L519—520: "One is the presence of optically thin cirrus clouds overlying opaque clouds,*

*which can only be detected by CALIOP. In such cloud vertical structures, the CTH and CBH retrieved by our algorithm are expected to correspond to those of the opaque clouds"* *– won't this be a bit more complicated and you might end up with something in between? How far into the clouds does the TIR channel typically see?*

Answer: Typically, TIR can retrieve cirrus clouds with a COT of 0.1 or greater (e.g., Iwabuchi et al., 2016), whereas the CALIOP lidar can detect optically thinner cloud layers with optical thicknesses as small as about 0.01 (e.g., Winker et al., 2009). Notably, CALIOP observations have revealed that cirrus clouds with COT < 0.1 are relatively common. Based on these facts, we assumed the existence of vertical cloud layers detectable by CALIOP alone but transmissive to TIR radiation. To clarify these points, the text has been revised as below.

[Section 4.4; Lines 584 - 589]

"One is the presence of optically thin cirrus clouds overlying opaque clouds, which can be detected by CALIOP but are transmissive to TIR radiation. This arises because TIR typically retrieves cirrus clouds with COTs of 0.1 or greater (e.g., Iwabuchi et al., 2016), while the CALIOP lidar is capable of detecting optically thinner cloud layers with optical thicknesses as small as about 0.01 (e.g., Winker et al., 2009). In such vertical cloud structures, the CTH and CBH retrieved by our algorithm are expected to correspond to those of the opaque clouds"

- Iwabuchi, H., Saito, M., Tokoro, Y., Putri, N. S., and Sekiguchi, M.: Retrieval of radiative and microphysical properties of clouds from multispectral infrared measurements, Progress in Earth and Planetary Science, 3, 1–18, https://doi.org/10.1186/s40645-016-0108-3, 2016.
- Winker, D. M., Vaughan, M. A., Omar, A., Hu, Y., Powell, K. A., Liu, Z., Hunt, W. H., and Young, S. A.: Overview of the CALIPSO Mission and CALIOP Data Processing Algorithms, J Atmos Ocean Tech, 26, 2310–2323, https://doi.org/10.1175/2009jtecha1281.1, 2009.
- Dupont, J. -C., Haeffelin, M., Morille, Y., Noël, V., Keckhut, P., Winker, D., Comstock, J., Chervet, P., and Roblin, A.: Macrophysical and optical properties of midlatitude cirrus clouds from four ground-based lidars and collocated CALIOP observations, J. Geophys. Res.: Atmos., 115, https://doi.org/10.1029/2009jd011943, 2010.
- Pandit, A. K., Gadhavi, H. S., Ratnam, M. V., Raghunath, K., Rao, S. V. B., and Jayaraman, A.: Long-term trend analysis and climatology of tropical cirrus clouds

using 16 years of lidar data set over Southern India, Atmos. Chem. Phys., 15, 13833–13848, https://doi.org/10.5194/acp-15-13833-2015, 2015.

---

## Author Comment (AC5)

First and foremost, we would like to express our sincere gratitude to Luca Lelli, the anonymous reviewers, the editor, and the editorial support team for taking the time to review our manuscript and provide valuable feedback. The comments we received were extremely helpful in improving our manuscript, and we are very grateful for them. As outlined below, we have revised the manuscript based on the feedback. The reviewers' comments are copied below and shown in *italics*, while our responses and the corresponding text in the manuscript are shown in red and orange, respectively.

**Response to the editorial support team**

*Regarding figures 3, 7: please ensure that the colour schemes used in your maps and charts allow readers with colour vision deficiencies to correctly interpret your findings. Please check your figures using the Coblis – Color Blindness Simulator (https://www.color-blindness.com/coblis-color-blindness-simulator/) and revise the colour schemes accordingly with the next file upload request.*

Answer: In response to the comment, we updated the color scheme for Figures 3 and 4 (excluding Figure 3a) to the 'Scientific Color Maps' recommended on the AMT submission page (https://www.atmospheric-measurement-techniques.net/submission.html).We recognize that adjusting the color scheme of the RGB images in Figures 3a and 7 as well would also be preferable. However, since the values of the three channels are directly assigned to R, G, and B, we are unsure how to modify them to make them colorblind-friendly. Instead, we utilized the 'Coblis – Color Blindness Simulator' to confirm that the RGB images in Figures 3 and 7 can be correctly interpreted by readers with anomalous trichromacy.

**Response to Anonymous Referee #5**

*This is a well-written comprehensive paper, presenting the use of O2-A band information for CBH/CGT and multiple cloud retrievals, which would be applicable to a few upcoming sensors. Below are just minor corrections and additions for clarification which would help improve the manuscript.*

We would like to thank you very much for carefully reading our manuscript and providing us with valuable comments. We have revised our manuscript, by taking full account of the referee's suggestions. The original comments are copied below and shown in *italics*, while our responses and the corresponding text in the manuscript are shown in red and orange, respectively.

*Suggestions/comments:*
*Evaluations using ground, ship, and space-based observations are very impressive. Just for more completeness of the paper, especially for the regional or global analysis cases, intercomparisons with other satellite sensor cloud data like MODIS or Himawari (even a visual inspection on the general patterns) would be wonderful here, rather than just addressing the consistency or well-known cloud regimes with the sole SGLI products.*
*Section 5.2: I thank the authors for including this section which contains many answers about my questions and thoughts already. Looking forward to seeing your next research outcomes based on these perspectives.*

Answer: As detailed in Section 2.1, the retrieval algorithm developed in this study is a coupling of two algorithms: (i) the four-channel algorithm (Kuji and Nakajima, 2001) using the VNIR, SWIR, TIR, and oxygen A-band channels, and (ii) a cloud phase retrieval algorithm using multiple SWIR channels (Nagao and Suzuki, 2021). As demonstrated in Nagao and Suzuki (2021), the algorithm of (ii) has already been evaluated through comparisons with MODIS and CALIOP. Therefore, this study focuses on the validation of CTH and CBH retrieved using the technique in (i) and compared these with measurements from ground-based and ship-borne ceilometers, as well as space-borne instruments, namely CALIOP and CloudSat/CPR, which are capable of measuring these cloud geometrical parameters. However, we agree that it is important to compare the cloud properties retrieved by our algorithm with the widely used products from MODIS and Himawari/AHI to enhance the reliability of our algorithm and to cross validate the cloud geometrical parameters inferred from multiple satellite measurements with passive

sensors. We will address this in future work.

***Minor comments:***

*Line 50: It would be good to add "narrow" before nadir*

Answer: We have added "narrow" and revised the sentence as follows:
[Section 1; Lines 49 - 50]
"Another limitation of these active sensors is that their measurements are constrained to narrow nadir views along the satellite's orbit."

*Line 77-78: Remove CBH and CGT here from other fundamental properties*

Answer: We have removed "CBH" and "CGT" and revised the sentence as follows:
[Section 1; Lines 85 - 87]
"Additionally, passive instruments designed for cloud remote sensing typically have multi-wavelength channels, allowing for the retrieval of other fundamental cloud properties, such as COT and CER"

*Line 125: It is not very clear. Please rework on this sentence.*

Answer: To clarify, we have revised the paragraph containing the sentence you mentioned as follows:
[Section 2.2.1; Lines 133 - 140]
"In the oxygen absorption channel, sunlight is significantly absorbed by the oxygen above the clouds before and after being reflected by the clouds on its path to the satellite. The TOA reflectivity in the oxygen absorption channel can be expressed with two additional parameters: CTH and the amount of oxygen above CTH. Conveniently, oxygen is well-mixed in the atmosphere, and its mixing ratio can be assumed to be globally constant and known. Thus, if the CTH (or cloud top pressure, equivalently) is given, the amount of oxygen above cloud can be immediately calculated. Therefore, when CGT~0, it is sufficient for parameterizing the TOA reflectance in the oxygen absorption channel to have CTH in addition to COT, CER, and cloud thermodynamic phase."

*Figure 2 caption: TH0, TH1, FAI in Fig. 2 caption have ever defined somewhere?*

Answer: We have added the explanations for these notations and revised the caption of Figure 2 as follows:
[Figure 2]
"…; $\theta_0$, solar zenith angle; $\theta_1$, sensor zenith angle; $\phi$, relative azimuth angle."

*Table 1: How to come up with Table 1 values?*

Answer: The values in Table 1 were provided roughly based on cloud property products from other satellite observations, without overly constraining the solution space. For the optimal estimation method to be most effective, a prior distribution close to the true value should be used. However, since this is the first application of our algorithm to SGLI, we used a normal distribution with means of typical orders of magnitude and fairly large standard deviations to avoid excessive reliance on the prior distribution. To clarify these points, the following text has been added.
[Section 2.2.1; Lines 221 - 225]
"Note that the values in Table 1 could be assigned more appropriate prior distributions (mean, standard deviation, and even covariance) by using cloud property products from other satellite observations. However, since this is the first application of our algorithm to GCOM-C/SGLI, we used a normal distribution for simplicity with means of typical orders of magnitude and fairly large standard deviations to avoid excessive reliance on the prior distribution."

*Line 206-207: This sentence leads to ask why for "rather than CBH". Just a slight revision could be done for clarification.*

Answer: We revised the sentence as follows:
[Section 2.2.2; Lines 241 - 243]
"For VN9, we introduced cloud base pressure, which is more directly related to the amount of oxygen within clouds compared to CBH, as a new variable to account for oxygen absorption within clouds."

*Line 208: Does it mean that this TIR region was missing in the original RTM?*

Answer: Yes, the retrieval algorithm used in this study is an updated version from the cloud phase retrieval algorithm of Nagao and Suzuki (2021), which only uses wavelengths from the visible to shortwave infrared region and thus does not handle the TIR region. To retrieve CTH using TIR channels, this study additionally implemented radiative transfer computations in the TIR region, based on techniques from previous studies (e.g., Nakajima and Nakajima, 1995; Kawamoto et al., 2001).

*Line 218: "TOA radiance product" means SGLI operational L1 data? Reference or data link?*

Answer: The precise answer is "No". The GCOM-C mission provides Level 1B scene radiance products, but the radiance product used in this study is the "LTOA" radiance product defined as Level 2, which provides radiance data projected onto a sinusoidal tile. The data link is described in the "*Data availability*" section as follows:

"*Data availability.* The GCOM-C/SGLI products for top-of-atmosphere radiation, cloud flags, cloud properties, and land surface reflectance, referred to as LTOA, CLFG, CLPR, and RSRF products, respectively, are available online on the Globe Portal System (G-Portal) of JAXA (https://gportal.jaxa.jp)."

*Line 218: "SGLI-measured refelctances and radiances -> Information specifically for which channels will be helpful.*

Answer: The SGLI LTOA radiance product naturally includes top-of-atmosphere (TOA) radiance for all the SGLI spectral channels. On the other hand, the channels used in the retrieval algorithm for this study are described in Section 2.2.1 as follows:
[Section 2.2.1]
"In the analysis in Sects 3 and 4, we employed the seven SGLI channels, VN9 (763 nm), VN11 (868 nm), SW1 (1.05 μm), SW3 (1.63 μm), SW4 (2.21 μm), TI1 (10.8 μm), and TI2 (12.0 μm), to retrieve the five cloud properties."

*Line 226-227: could you add a little bit more details about how to correct it, not just*

*added flags.*

Answer: We have rephrased it as follows:

[Section 2.3; Lines 264 - 266]

"Notably, we effectively corrected for the impact of land and sea surface reflectance on the SGLI observed radiances in the manner described above. However, our algorithm did not explicitly account for the presence of sea ice over the ocean or its high reflectance."

*Line 232: Add the MERRA-2 data source here or to the data availability section at the end.*

Answer: We have added the following text in the Data availability section:

[Data availability]

"The Modern-Era Retrospective Analysis for Research and Applications Version 2 (MERRA-2) product are available online through the Goddard Earth Sciences Data and Information Services Center (http://disc.sci.gsfc.nasa.gov/mdisc/)."

*Figure 3: Add the time to Fig. 3 caption*

Answer: We have revised the caption as follows:

[Figure 3]

"…, observed at around 01:15 UTC (around 10:30 AM local sun time) on October 1, 2021."

*Line 244: add 'RGB' color composite*

Answer: We have added 'RGB' as follows:

[Section 3.1; Lines 288 - 289]

"Figure 3a shows an RGB color composite image"

*Line 385: I understand the difficulties to obtain matchup data, but still 30 min average seems like quite a relaxed threshold.*

Answer: To investigate the point the reviewer raised, Figure S3 is newly added and included in the supplemental material. This illustrates how the bias, RMSE, and correlation between CBHs from SGLI and ceilometer depend on ($\Delta s$, $\Delta t$). This figure demonstrates that the bias and RMSE worsen when $\Delta t$ is set to less than 30 minutes. Additionally, it shows that the choice of ($\Delta s$ < 4 km, $\Delta t$ < 30 min) is not unique to minimize the bias and error; in other words, there are other combinations of values that may yield better agreement in CBH between SGLI and ceilometer.

Therefore, the following text have been added to the main text as well:

[Section 4.2; Lines 467 - 472]

"It should be emphasized that the thresholds ($\Delta s$, $\Delta t$) can influence the results in Fig. 6, but are not critical. Figure S3 in the supplementary material illustrates the dependence of bias, RMSE, and correlation coefficient between CBHs from SGLI and ceilometers on ($\Delta s$, $\Delta t$). It demonstrates that bias and RMSE worsen when $\Delta t$ is set to shorter than 30 min. Additionally, it indicates that the choice of $\Delta s$ < 4 km and $\Delta t$ < 30 min is not the only method to minimize bias and error. In other words, there are alternative values for ($\Delta s$, $\Delta t$) that can yield better agreement in CBH between SGLI and ceilometer.
"

*Line 507: remove "," after SGLI*

Answer: We have removed ",".

*Line 543: Maybe "Moreover" would be better, if the authors intended to address it is good to be sensitive to other cloud properties.*

Answer: In accordance with the comment, the term "However" has been replaced with "Moreover" as follows:

"[Section 5.1; Lines 619 - 621]

The 763 nm channel, located within the oxygen A-band, can provide CBH and CGT through satellite-based passive remote sensing. Moreover, the challenge in utilizing the 763 nm channel is that it is sensitive not only to CBH and CGT but also to CTH and other cloud properties."

*Line 566-567: "This underestimation of the CTH also suggests a systematic underestimation of the CGT by the SGLI. " Any suggestions or further thoughts for this?*

Answer: The underestimation of the SGLI CGT was primarily attributed to the underestimation of CTH retrieved, rather than to errors in CBH retrieval. Meanwhile, the SGLI CTH showed good agreement with the MOD06 CTH, which is similarly derived from TIR measurements. This result underscores that the well-known and persistent issue of TIR-derived CTHs being systematically lower than those detected by CALIOP plays a critical role in CGT estimation when combining the 763 nm and TIR channels. To provide additional context, the following text has been added at the end of the paragraph containing the sentence you mentioned:

[Section 5.1; Lines 646 - 648]

"This result highlights the well-known and persistent issue that CTHs derived from TIR are systematically lower than those detected by CALIOP, underscoring its critical impact on CGT estimation when combining the 763 nm and TIR channels."

*Line 596: the "current" CTH retrieval -> just for clarification*

Answer: We have added "current" as follows:

[Section 5.2; Lines 674 - 675]

", and additional TIR channels cannot be added to improve the current CTH retrieval."

---

## Author Comment (AC6)

First and foremost, we would like to express our sincere gratitude to Luca Lelli, the anonymous reviewers, the editor, and the editorial support team for taking the time to review our manuscript and provide valuable feedback. The comments we received were extremely helpful in improving our manuscript, and we are very grateful for them. As outlined below, we have revised the manuscript based on the feedback. The reviewers' comments are copied below and shown in *italics*, while our responses and the corresponding text in the manuscript are shown in red and orange, respectively.

**Response to the editorial support team**

*Regarding figures 3, 7: please ensure that the colour schemes used in your maps and charts allow readers with colour vision deficiencies to correctly interpret your findings. Please check your figures using the Coblis – Color Blindness Simulator (https://www.color-blindness.com/coblis-color-blindness-simulator/) and revise the colour schemes accordingly with the next file upload request.*

Answer: In response to the comment, we updated the color scheme for Figures 3 and 4 (excluding Figure 3a) to the 'Scientific Color Maps' recommended on the AMT submission page (https://www.atmospheric-measurement-techniques.net/submission.html).We recognize that adjusting the color scheme of the RGB images in Figures 3a and 7 as well would also be preferable. However, since the values of the three channels are directly assigned to R, G, and B, we are unsure how to modify them to make them colorblind-friendly. Instead, we utilized the 'Coblis – Color Blindness Simulator' to confirm that the RGB images in Figures 3 and 7 can be correctly interpreted by readers with anomalous trichromacy.

*GENERAL COMMENTS*

*SGLI onboard GCOM-C is a powerful instrument to cover wide spectral range of both solar reflected light and thermal emission. By adding O₂A information, understanding vertical distribution of clouds will be much improved. Authors referred their former studies. However, the description of why several cloud parameters such as base height and thickness can be retrieved from space is essential. How many parameters can be retrieved by assuming how many parameters from how many spectral channels should be described. In addition, the degree of freedom and uncertainties for each retrieved parameter using the optimal estimation method should be presented. I recommend major revision before its AMT publication.*

Answer: We would like to thank you very much for carefully reading our manuscript and providing us with valuable comments. We have revised our manuscript, by taking full account of the referee's suggestions. The original comments are copied below and shown in *italics*, while our responses and the corresponding text in the manuscript are shown in red and orange, respectively.

*I have the following general questions.*

*(1) SGLI covers the O₂A band with one spectral channel, of which spectral radiance depends on observation geometry, cloud height and fraction, surface albedo etc. The algorithm cannot use individual lines within the O₂A band. Large airmass causes saturation in strong absorption lines. Do viewing geometry and solar zenith angle affect the quality of retrieval? If uncertainties from retrieved cloud parameters varies with latitude etc., it should be presented.*

Answer: In response to this comment, we have included a new figure, Figure S5, in the revised supplemental material, illustrating the angular dependence of cloud properties. Figure S5 presents the zonal means of a) COT, b) CER, c) ICOTF, d) CTH, and e) CBH, calculated for specific ranges of satellite zenith angles. Figure S5 shows that the deviation of zonal means of all these parameters with respect to the satellite zenith angle is relatively small compared to the zonal mean values themselves, indicating that the angular dependence of our algorithm is not significant. However, the zonal mean of CBH shows a moderate variation with the satellite zenith angle, particularly in low-latitude regions. This may be due to the fact that a larger satellite zenith angle causes the sensor

to observe the side view of tall clouds, such as deep convective clouds. It remains crucial to continue investigating retrieval uncertainties related to the three-dimensional structure of clouds.

Therefore, the following text have been added to the main text as well:

[Last paragraph of Section 5.2; Lines 687 - 695]

"The supplementary material also presents additional analysis results that may help in examining the issues that remain in our retrieval algorithm. Figure S5 demonstrates that the deviations in the zonal mean for all cloud properties retrieved by our algorithm were relatively minor with respect to the satellite zenith angle, indicating that the angle dependence of the algorithm is not significantly influential. However, the moderate angle-dependent variations in the zonal mean of CBH at low latitudes may require further investigation."

*(2) Forward calculation: retrieved parameters must be defined in the forward model. Definition of vertical layers, cloud top and bottom height, optical thickness of high-altitude cirrus cloud and aerosol help readers' understanding. Which cloud-related parameters are retrieved, and which are assumed? Do authors assume a single pixel is fully covered? Do they consider popcorn like clouds?*

Answer: In the revision, we have added a new supplemental material containing the details of the forward model used in our retrieval algorithm. The forward model can consider 12 variables listed in Table S1 and surface albedo $A_s$ in Equation (A2). Of these variables, five variables for cloud properties ($\tau_c$, $r_e$, $ICOTF$, $P_c$, and $P_b$) were estimated by the inversion process, three variable for solar-sensor geometry $(\theta_0$, $\theta_1$, and $\phi)$ were given from the SGLI observations, and four variables for atmospheric condition ($P_S$, $TPW$, $COL$, and $DU$) were given from using the MERRA-2 products. In addition, $A_s$ for land was estimated from the SGLI land reflectance product and $A_s$ for ocean was estimated from sea surface wind speeds. To include an explanation of the supplemental material, we have added the following texts in Section 2.2.2.

[Section 2.2.2; Lines 245 - 246]

"The technical details of the forward model are provided in the supplemental material (see Text S1)."

In contrast, the use of the MERRA-2 and SGLI land surface products to determine atmospheric and surface conditions is described in Section 2.3 in the original manuscript

as follows. Note that the mathematical symbols used in the supplement text S1 are not used to avoid complications in the explanation:

[Section 2.3; Lines 259 - 264]

"The third input was the SGLI land surface reflectance product (referred to as SGLI-RSRF). It provides land surface reflectance for the VNIR-to-SWIR channels of the SGLI, along with the parameters for the bidirectional reflectance distribution function model, which are input into the forward model for land pixels. For ocean pixels, the RSTAR7 subroutine was employed to estimate sea surface reflectivity from sun–satellite geometry and sea surface wind speed, which were then fed into the forward model."

*(3) By assuming the signal to noise ratio of SGLI and other uncertainties such as none-linearity of electronics, radiometric calibration error, what is the expected detection limit or uncertainties of these parameters from theoretical optimal estimation method? The values of a prior distribution and ranges are well summarized in Table 1. How about posterior? What are the results using real SGLI data versus posterior? These descriptions will improve readers' understanding of the validation part of this paper.*

Answer: In the supplemental material, we have included a section (Text S2) on sensitivity analysis based on error propagation theory and radiative transfer simulation, together with the results shown in Figures S1 and S2. Figure S1 demonstrates how perturbations in SW1, SW4, SW3, TI1, and VN9 propagate to the retrievals of COT, CER, ICOTF, CTH, and CBH. Figure S2 illustrates how uncertainties in the measurement vector induce uncertainties in the retrievals. Figures S2a and S2b compare different combinations of channels used in the retrieval process.

We agree that this sensitivity analysis is useful for understanding the behavior of our algorithm; however, we would prefer to limit it to the supplemental material. This is because the comparison with the ceilometer and other satellites is the main focus of this study, which reveals realistic error factors (such as the vertical inhomogeneity of cloud characteristics and multilayer cloud structure) that are difficult to incorporate into the sensitivity analysis.

Therefore, the following text have been added to the main text as well:

[Last paragraph of Section 4.1; Lines 412 - 425]

"In our algorithm, the uncertainty in CBH retrieval is also entangled with the uncertainty in COT retrieval. We performed a sensitivity analysis based on the error propagation theory to examine how measurement uncertainties propagate to retrieval uncertainties

(see Text S2 in the supplemental material). Figure S1 demonstrates how perturbations in individual measurement channels induce retrieval errors. Notably, perturbations in SW1, which is a channel sensitive to COT but located outside the oxygen A-band, can induce errors not only in COT retrieval (Fig. S1(1,1)) but also in CBH retrieval (Fig. S1(5,1)). This indicates that COT errors disturb the separation of COT and CBH from VN9 measurements. Figure S2 further demonstrates how the overall uncertainty in the multi-wavelength measurements incorporated into the inverse estimation propagates to retrieval uncertainties. The comparison of Figs. S2a1 and S2b1 reveals incorporating VN11 alongside SW1 reduce the uncertainty in COT retrieval, which, in turn, contributes to reduce uncertainty in CBH retrieval. As described in Section 2.2.1, our algorithm utilized both SW1 and VN11. The results of these sensitivity analyses emphasize the importance of carefully addressing uncertainties in COT retrieval when deriving CBH from VN9 measurements. The entanglement of COT, CTH, and CBH retrieval errors associated with oxygen A-band measurements has also been reported by Lelli et al. (2014)."

*(4) For the last 10 years, line parameters of the $O_2$ A band have been much improved by innovative laboratory spectroscopy. Which database the authors used? Do authors use line by line calculation for the $O_2A$ band or look up tables in their forward model?*

Answer: The RSTAR package used for radiative transfer calculations in this study contains gas absorption tables compiled using the k-distribution method for narrowband channels, provided with a spectral resolution of approximately 0.8 nm in the oxygen A-band region. Radiative transfer calculations are performed for each narrowband channel and subsequently integrated using the spectral response function of the SGLI VN9 channel to simulate TOA radiances. However, the gas absorption line database underpinning this k-distribution table is HITRAN2004. As you have noted, this means that recent updates to oxygen absorption line data are not incorporated. To clarify this point, we have added the following text:
[Section 2.2.2; Lines 234 - 237]
"The RSTAR version 7 (RSTAR7) package includes gas absorption line tables compiled into narrow bands using the k-distribution method. However, the k-distribution table is based on the HITRAN 2004 molecular spectroscopic database (Rothmana et al., 2005) and does not incorporate recent updates to the oxygen absorption lines."

*(5) A priori information. How many A priori information such as aerosol type, surface pressure, wind speed over the ocean are included? How much uncertainties are assumed?*

Answer: As described in Section 2.3 of the main text, the atmospheric and surface variables other than cloud properties handled by the forward model include temperature profile, water vapor profile, surface pressure, surface temperature, and sea surface wind speed, which are given by the MERRA-2 product. In addition, land surface reflectance is given by the SGLI-RSRF, the operational land surface reflectance product of the GCOM-C/SGLI mission. These atmospheric and surface variables excluding cloud properties, are treated as known, and their uncertainties are not explicitly considered in the inverse estimation. Aerosols are neglected in both the forward model and inverse estimation. To clarify these points, the following text has been added to Section 2.3.
[Section 2.3; Lines 275 - 279]
"It should be noted that the uncertainties in the meteorological variables provided by the MERRA-2 reanalysis data, as well as those in the land surface reflectance data from the SGLI-RSRF, were not explicitly accounted for in the inverse estimation. Furthermore, the impacts of aerosols on the observed radiance were not considered in either the forward model or the inverse estimation."

*SPECIFIC COMMENTS*
*(1) Page 1, lines 21-22*
*What is the difference between "systematic" bias in line 20 and bias in line 21?*

Answer: "systematic" was an inappropriate adjective. In the revised manuscript it has been removed and simply written as 'bias'.
[Abstract; Lines 21 - 22]
"These include the bias of SGLI CTH related to cirrus clouds and the bias of SGLI CBH caused by multi-layer clouds."

*(2) Page 11, Line 267*
*What do authors mean by "negatively affect cloud retrieval"? Generally speaking, by properly considering uncertainties, adding spectral channel for retrieval provide information.*

Answer: The phrase included in the original text did not intend to mean "*negatively affect cloud retrieval*" but more precisely intended to mean "*negatively affect cloud phase retrieval*". In our previous study (Nagao and Suzuki, 2021), we developed a retrieval algorithm for cloud properties (COT and CER) and cloud thermodynamic phase (ICOTF) using the SWIR channels. In this study, we extended the algorithm to incorporate the TIR and oxygen A-band channels, enabling the simultaneous estimation of CTH and CBH. The sentence containing the phrase in question indicates that the quality of the original cloud phase retrieval using SWIR channels was not compromised by the functional extension to include the TIR and oxygen A-band channels. To clarify this point, the revised sentence is as follows.

[Section 3.1; Lines 310 - 312]

"In other words, the incorporation of the TIR and oxygen A-band channels in this study did not adversely impact the quality of cloud phase retrieval based on the SWIR channels."

*(3) Page 19, Line 433,*

*What are the definitions of mid- and high-level clouds? What is the difference from "lower-level" in line 363?*

Answer: The sentences that contain the phrases in question are as follows:

- "In contrast, for thin cirrus clouds, CTHs are underestimated by 2–3 km relative to those detected by CALIOP due to factors such as COT and overlap with lower-level clouds (Baum et al., 2012; King et al., 2013)" (Section 4.1)

- "The CBH retrievals using the SGLI oxygen A-band were most effective for mid- and high-level clouds." (Section 4.2)

The former sentence cites discussions from previous studies on CTH retrieval errors in the case of multi-layered clouds. In this context, the term "lower-level clouds" refers to clouds situated below cirrus clouds. On the other hand, the latter sentence describes the results in Figure 6, and the term "mid- and high-level clouds" is defined in the paragraph preceding it as "mid- and high-level clouds, as suggested by the relatively high CBH (> 4 km), and the examples shown in Fig. 7g–i".

*TECHNICAL CORRECTIONS*
*(1) Line 544, "CTT"*

*It appears first in this paper. It looks typo.*

Answer: "CTT" has been replaced by "CTH".
[Section 5.1; Lines 620 - 621]
"Moreover, the challenge in utilizing the 763 nm channel is that it is sensitive not only to CBH and CGT but also to CTH and other cloud properties."